# Global transcontinental power pools for low-carbon electricity

Haozhe Yang [1], Ranjit Deshmukh [1,2] & Sangwon Suh [1] ✉

The transition to low-carbon electricity is crucial for meeting global climate goals. However, given the uneven spatial distribution and temporal variability of renewable resources, balancing the supply and demand of electricity will be challenging when relying on close to 100% shares of renewable energy. Here, we use an electricity planning model with hourly supply-demand projections and high-resolution renewable resource maps, to examine whether transcontinental power pools reliably meet the growing global demand for renewable electricity and reduce the system cost. If all suitable sites for renewable energy are available for development, transcontinental trade in electricity reduces the annual system cost of electricity in 2050 by 5–52% across six transcontinental power pools compared to no electricity trade. Under land constraints, if only the global top 10% of suitable renewable energy sites are available, then without international trade, renewables are unable to meet 12% of global demand in 2050. Introducing transcontinental power pools with the same land constraints, however, enables renewables to meet 100% of future electricity demand, while also reducing costs by up to 23% across power pools. Our results highlight the benefits of expanding regional transmission networks in highly decarbonized but land-constrained future electricity systems.

To limit the global mean temperature increase relative to the pre-industrial era within 2 °C by 2100, carbon dioxide ($CO_2$) emissions from fossil fuels must approach zero by the middle of this century[1,2]. Among anthropogenic sources, the electricity sector contributes about 40% of the global energy-related $CO_2$ emissions annually[3]. In addition, electrifying other sectors including transport and industry is a predominant carbon mitigation strategy. It is, therefore, crucial to decarbonize the electricity sector, especially using abundant renewable resources[4]. In recent decades, the cost of renewable energy, especially solar and wind, has declined substantially[5–7]. The levelized cost of electricity generated by solar and wind has become competitive compared to the electricity generated by fossil fuels and nuclear[7]. The global share of wind and solar generation in total electricity supply has risen from <1% in 2000 to nearly 10% in 2020[6].

The global potential of renewable electricity is immense—solar, wind and hydropower electricity can supply ~135, ~840, and ~50 petawatt hours (PWh) of electricity a year, respectively, according to some estimates[8–11]. If land suitability and availability are considered, the global potential for renewable energy is reduced to 50–400 PWh per year[12], which is still ~2–17 times higher than the 23 PWh[13] of global electricity consumption in 2018. However, the temporal variability of renewable resources may limit their potential for reliably meeting electricity demand[14]. Without energy storage and over-generation (more electricity generation than demand), wind and solar may fulfill only 70–90% of the current electricity demand[15]. Furthermore, renewable resources are unevenly distributed across space[10,12,16], further exacerbating the problem of spatiotemporal mismatch between supply and demand. The uneven distribution of renewable resources also creates substantial variation in the cost of renewable electricity across countries and regions[12], undermining their cost-effectiveness, especially if renewable resources are developed to meet electricity demand only within national borders. Building regional power pools and increasing electricity trade can and has been pursued to address the spatiotemporal mismatch between renewable electricity

[1]Bren School of Environmental Science and Management, University of California, Santa Barbara, CA, USA. [2]Environmental Studies Program, University of California, Santa Barbara, CA, USA. ✉e-mail: suh@bren.ucsb.edu

generation and demand[17]. In fact, several regional power pools have been in operation in Europe, North America, and Southern Africa[18]. Existing intercontinental transmission projects include the BritNed (United Kingdom and Netherlands), NordBalt (Sweden and Lithuania), and NordNed (Norway and Netherlands)[19]. In 2020, cross-border trade of electricity accounts for 2.8% of the global supply[20].

Expanding regional power pools to continental-scale power pools can further increase electricity trade, decrease costs, and enable the integration of near-100% shares of renewable energy. Guo et al.[21] and Zhao et al.[22] examined the implications of a global power pool by modeling electricity trade between continents and found an increase in the share of renewable energy with increasing trade. However, these studies did not examine a near-100% clean energy system. Other studies, specifically from the Lappeenranta University of Technology, examined the trade of electricity between subregions across continental-scale power pools with 100% renewable electricity systems (e.g. Europe[23], Sub-saharan Africa[24], Northeast Asia[25], and MENA[26]), but these studies do not incorporate country-level spatial resolution. Further, none of these studies examined the impact of land-use constraints on renewable energy resource availability and its resulting international electricity trade and system cost implications.

Here, we model the investments and operations of 100% renewable electricity systems across 211 countries and administrative areas in 2050, to quantify the benefits in reliability and system cost through introducing transcontinental power pools, compared to the case without electricity trade. Renewable resources analyzed in this study include solar photovoltaics, concentrated solar power, onshore wind, offshore wind, and hydropower. We use their supply potentials at a $0.01° \times 0.01°$ spatial resolution across the globe. The supply potentials are then mapped with the demand balance using 2050 demand projections and an electricity-system planning model at an hourly resolution. We map the supply potential and demand at hourly intervals, first, within each country (country scenario) and, second, across transcontinental power pools (transcontinental scenario); both with only renewables.

Using the electricity system model, we co-optimize the investment and operation of electricity generation, transmission, and storage using a 3-h temporal resolution in 365 days of a whole year. To examine the reliability of the system across all hours of the year, we fix the optimized capacities and rerun the electricity system model to simulate the hourly operations across all 365 days. The demand projection follows the Sustainable Development Goal (SDG scenario) by IEA[27]. We assume a penalty of $100 million per MWh for the loss of load and a 1.6% transmission loss per 1000 kilometers[28]. We assume that the loss of load in the modeled 100% renewable energy system is exogenously satisfied by fossil fuels, and that the system cost of electricity comprises the cost of renewables, fossil fuels, and climate costs set at the social cost of carbon[29].

Under the transcontinental power pool scenario, we assume that countries within a continental region engage in electricity trade and share generation resources to meet their local electricity demand. We use six continental regions based on the current structure of global electricity trade[30] and proposed regional electricity networks[23,25,31–34]. These six regions are: Sub-Saharan Africa, East Asia and Russia (East Asia, South Asia, Central Asia, and Russia), Europe and MENA (Europe, Middle East, and North Africa), North America, South America, and Southeast Asia and Oceania.

To understand the implications of land constraints on renewable energy potential, costs, and benefits of transcontinental power pools, we examine two cases of land availability for renewable energy siting. First, all suitable sites for renewables are available for development. However, not all potential renewable resources can be tapped for electricity generation due to constraints on land availability not captured by available geospatial datasets, including ecological impacts, market accessibility, and local political support[16,35]. Therefore, in the

second case, for each renewable energy technology (excluding rooftop PV), we assume that only the top 10% of suitable sites at the global level[16] are available for energy development. We rank and select the top 10% renewable resources based on a composite index of resource yield (annual capacity factor), land use, infrastructure, and market accessibility (see Methods section).

## Results

### Mismatch between renewable resources and electricity demand
Globally, with all suitable land for renewable resources, the renewable potential reaches ~3500 PWh. If each country sources renewable resources within their national border to meet their national demand, renewable resources can supply 42 PWh of global electricity demand, but are 0.8 PWh or 2% short of the global annual demand in 2050 (Fig. 1a).

We find that renewables alone reliably meet 100% of electricity demand in three quarters of the countries. However, they fall short in meeting hourly demand in several countries (Fig. 1c). Notably, in South Korea, the renewable potential (0.6 PWh) is lower than the demand (1.1 PWh) by 0.5 PWh, leading to over 30% of unmet demand. In Switzerland, although the renewable potential (0.14 PWh) exceeds the annual demand (0.08 PWh), 0.01 PWh or 16% of the demand is not met by renewables alone. This is because the battery storage only balance the diurnal variability of renewable energy, and cannot balance the seasonal variation.

If only the top 10% suitable land is used, the global renewable potential is reduced by 80% to 665 PWh. Globally, 4.1 PWh of demand shortage is solely caused by a lack of aggregate renewable resources within the country boundary (Supplementary Fig. 1). After considering the temporal mismatch between renewable energy generation and demand, the renewable potential is 5.2 PWh or 12% short of annual electricity demand, if sourced within each national border (Fig. 1b). Because of the temporal variation in renewable energy generation, some countries cannot meet their demand by only renewables even if their annual renewable potential exceeds demand. For example, India is unable to meet ~11% of its demand with renewables alone because the renewable potential is only 1% higher than the demand. On the other hand, abundant resources are unutilized in countries with rich endowments of renewables. In China, for example, about one-third of the renewable potential is not utilized, if consumed only within the national border. In the United States, Brazil and Saudi Arabia, the potential of renewable resources is higher than the annual demand by an order of magnitude.

Under the constrained land availability scenario, in 2050, because of the uneven spatial distribution and the temporal variability of renewable resources, renewable electricity generation alone is unable to reliably meet over 20% of electricity demand in nearly one-third of the countries (Fig. 1d), mainly in Southeast Asia and East Asia. In many South European countries, over 10% of demand is unmet by renewables. Across North America, South America, Western Asia, Africa, and China, Mongolia and Russia, however, <3% of demand was unmet by renewables. Adding existing inter-country transmission lines reduces the gap in demand and renewable energy supply because of electricity trade (Supplementary Table 1), e.g., unmet demand in southern European countries decreases from 10–20% to 5–10% (Supplementary Fig. 2).

### Large variation in system cost of electricity
Due to the heterogeneity in renewable resource quality and abundance, we find that the system costs of electricity vary greatly across countries under the country scenario (Fig. 2a, b). These costs vary both across and within the six regions that we define for forming the transcontinental power pools (Fig. 2c). By utilizing all suitable sites, the country-level system costs of electricity are highest among countries in Europe, and East and Southeast Asia, exceeding $60/MWh. The lowest

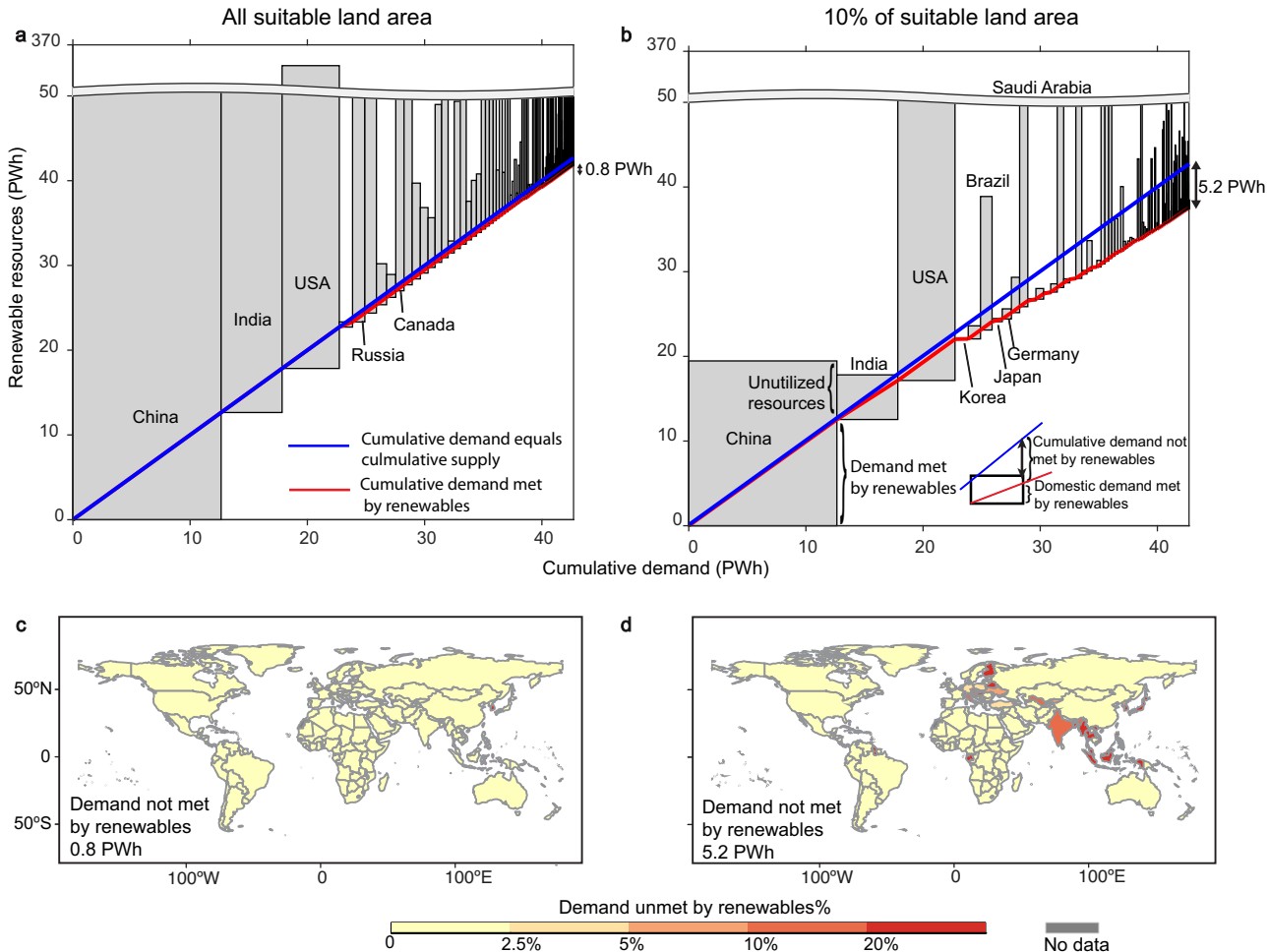

**Fig. 1 | Supply and demand of renewable electricity in 2050.** Renewable generation and electricity demand in 2050 by country under the 'country' scenario, assuming **a** all suitable sites for renewables and **b** top 10% suitable sites at the global level. Unmet demand with only supply of renewable resources under the 'country' scenario, assuming **c** all suitable sites for renewables and **d** top 10% suitable sites at the global level. In **a** and **b**, each rectangle represents the demand for electricity (horizontal dimension) and the available renewable resources potential (vertical dimension). The blue diagonal line represents demand equals renewable electricity generation by country (y = x line). The red line represents the cumulative demand met by renewable electricity generation by country. Within a country's rectangle, when the slope of the red line is smaller than the slope of the blue line, country-level electricity demand remains unmet using only renewable energy supply.

system cost occurs in South America, North America, Sub-Saharan Africa and Oceania, estimated to be around $30/MWh.

In Europe and MENA, East Asia and Russia, and Southeast Asia and Oceania, countries with large unutilized renewable potential have lower system costs (Fig. 2d). Within Europe and MENA, Middle Eastern and North African countries have over 20 PWh of excess renewable resources, and their system costs are much lower than those in Europe (Fig. 2a). Within the East Asia and Russia region, Russia, Pakistan and Afghanistan have over 10 PWh of excess renewable resources, with system costs (~$35/MWh) much lower than the average system cost in the region ($57/MWh). In Southeast Asia and Oceania, countries in Oceania have excess renewable resources and lower system costs, while Southeast Asian countries have higher costs or less renewable resources.

Limiting the available renewable resources to the global top 10% suitable sites substantially increases the system costs of electricity in countries with fewer renewable resources compared to their electricity demand (Fig. 2e). Within Southeast Asia and Oceania, 70% of the countries (mostly in Southeast Asia) suffer from a shortage of renewable resources, resulting in system costs of over $70/MWh. Only in countries such as Australia and Cambodia that have large renewable resources relative to their demand, system costs remain low (~$50/MWh). In East Asia and Russia, Pakistan, Afghanistan, and Russia

are the only three countries with over 0.5 PWh of excess renewable resources relative to demand and low system costs (~$40/MWh), whereas over half of the countries in the region have system costs over $60/MWh. In Europe and MENA and Sub-Saharan Africa regions, over half of the countries have system costs over $60/MWh and $50/MWh, respectively, because of limited renewable resources compared to demand.

In contrast, some regions and countries do not see a substantial increase in system costs under constraints on suitable sites. Because the top 10% suitable sites across South America and North America include abundant high-quality renewable resources to meet the electricity demand, system costs remain low. Similarly, countries that have high-quality renewable resources, such as wind resources in the United Kingdom, can access the same resources as the scenario without land constraints and as such, do not experience much system cost increases. Lastly, the average system cost in Europe and MENA see a decrease under a land constraint scenario. When 10% of the suitable sites are available, 0.4 PWh (6%) of the demand is exogenously met by fossil fuels, which is ~120 $/MWh. When all suitable sites are used, over 99% of electricity demand is met by renewables alone by building over capacities of solar and wind. Even if adding the fossil fuel cost, the average system cost with 10% suitable sites is lower than the average system cost with all suitable sites (Fig. 3).

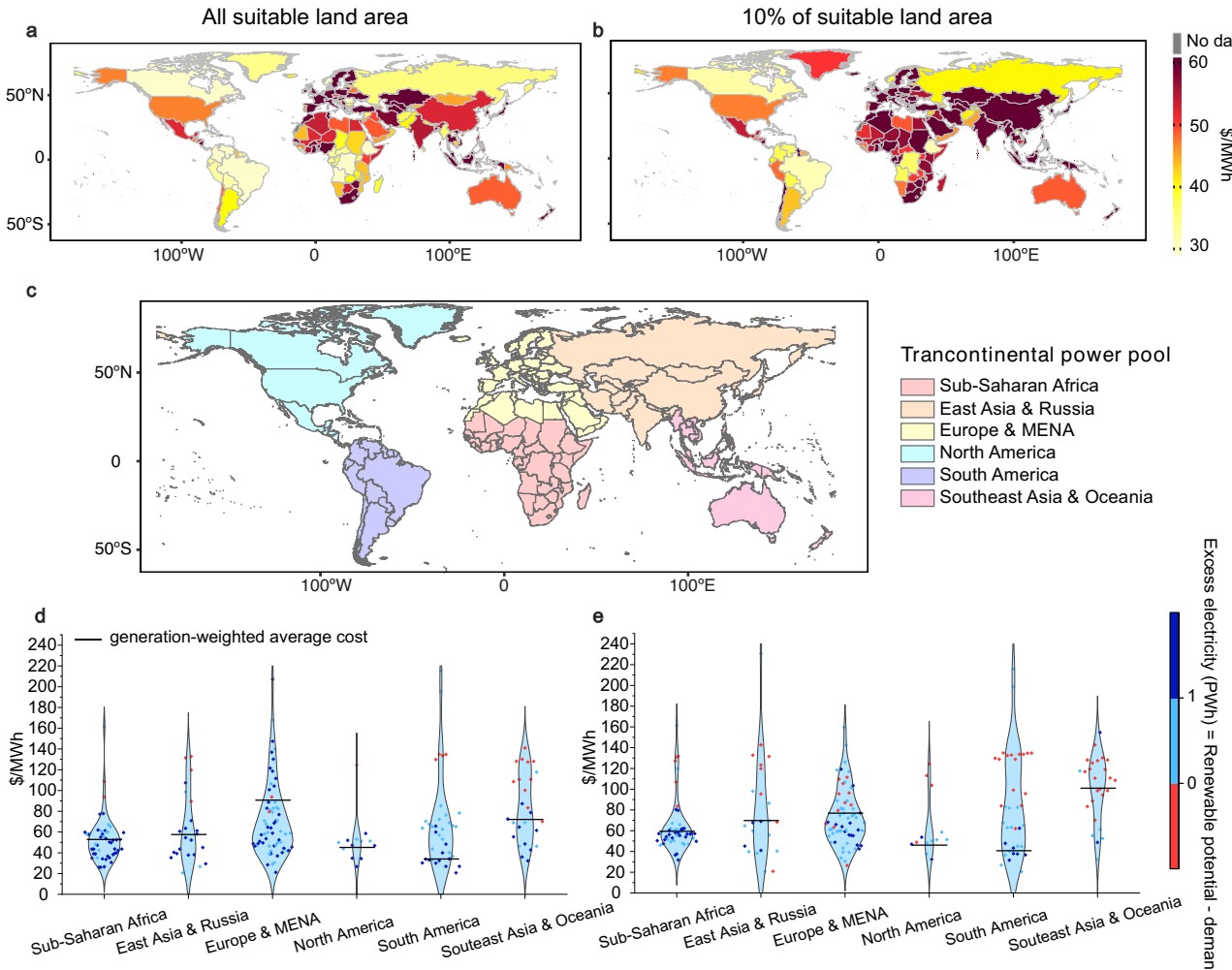

**Fig. 2 | The system cost of electricity (cost per unit of electricity demand) under the 'country' scenario.** The system cost of electricity under the 'country' scenario assuming **a** all suitable sites for renewables, and **b** only the global top 10% suitable sites for renewables are available for development. **c** Classification of the transcontinental power pools. Distribution of system costs across countries within the six regions assuming **d** all suitable sites for renewables, and **e** only the global top 10% suitable sites for renewables are available for development. In **d** and **e**, the color of scatters represents renewable energy potential excess or deficit (renewable energy potential minus electricity demand) in 2050.

## Transcontinental power pools avoid shortages and reduce costs

Under the transcontinental scenario, countries with an excess of renewable resources export their renewable electricity to countries with insufficient renewable resources. By utilizing all suitable sites for renewable resources, regional renewable resources are larger than their electricity demand by 27–1000 times (Fig. 3a). In Sub-Saharan Africa, the renewable resources potential (926 PWh) is nearly 1000 times the 2050 electricity demand (0.9 PWh). In East Asia and Russia where the regional demand is the largest across all regions at 23 PWh, the renewable resources potential is 628 PWh or 27 times the electricity demand. By integrating the regional renewable resources through transcontinental power pools, globally, the unmet demand by renewables decreases from 0.8 PWh (2%) to 0.0 PWh.

Compared to the country scenario, transcontinental power pools decrease system costs of electricity by 12–52% across all regions except for North America (Fig. 3c). Cost reductions in North America are small (5%) because of abundant renewable resources in each of the member countries. The largest reductions in system cost resulting from transcontinental power pools occur in Europe and MENA, with an average reduction of 52% or $47/MWh across all countries (Fig. 3c). These cost reductions are largely driven by the international trade of electricity requiring fewer capacities of renewables and battery storage. In

Europe and MENA, the decrease in cost for PV is the largest at $24/MWh, followed by the onshore wind ($10/MWh) and storage ($8/MWh). Assuming existing inter-country transmission lines in the Europe and MENA region in the country scenario reduces the system cost reduction benefits of a transcontinental power pool to 46% (Supplementary Table 2).

Cost reductions are also substantial in other power pools compared to the country scenario—over 10% in Sub-Saharan Africa, East Asia and Russia, South America, and Southeast Asia and Oceania. The cost reduction in these power pools is mainly driven by fewer installed PV and battery storage capacities in the transcontinental scenario.

By restricting renewable energy development to only the top 10% of suitable sites, regional renewable resources are still greater than the regional electricity demand in 2050 (Fig. 3b). Specifically, in Southeast Asia and Oceania and East Asia and Russia, renewable resources are more than tenfold and threefold of the demand in power pools, respectively. Globally, if renewable resources are shared within the six continental regions defined in this study, the unmet demand decreases from 5.2 PWh (12%) in the country scenario to 0.0 PWh, which is similar to the unmet demand if all suitable renewable energy sites were developed.

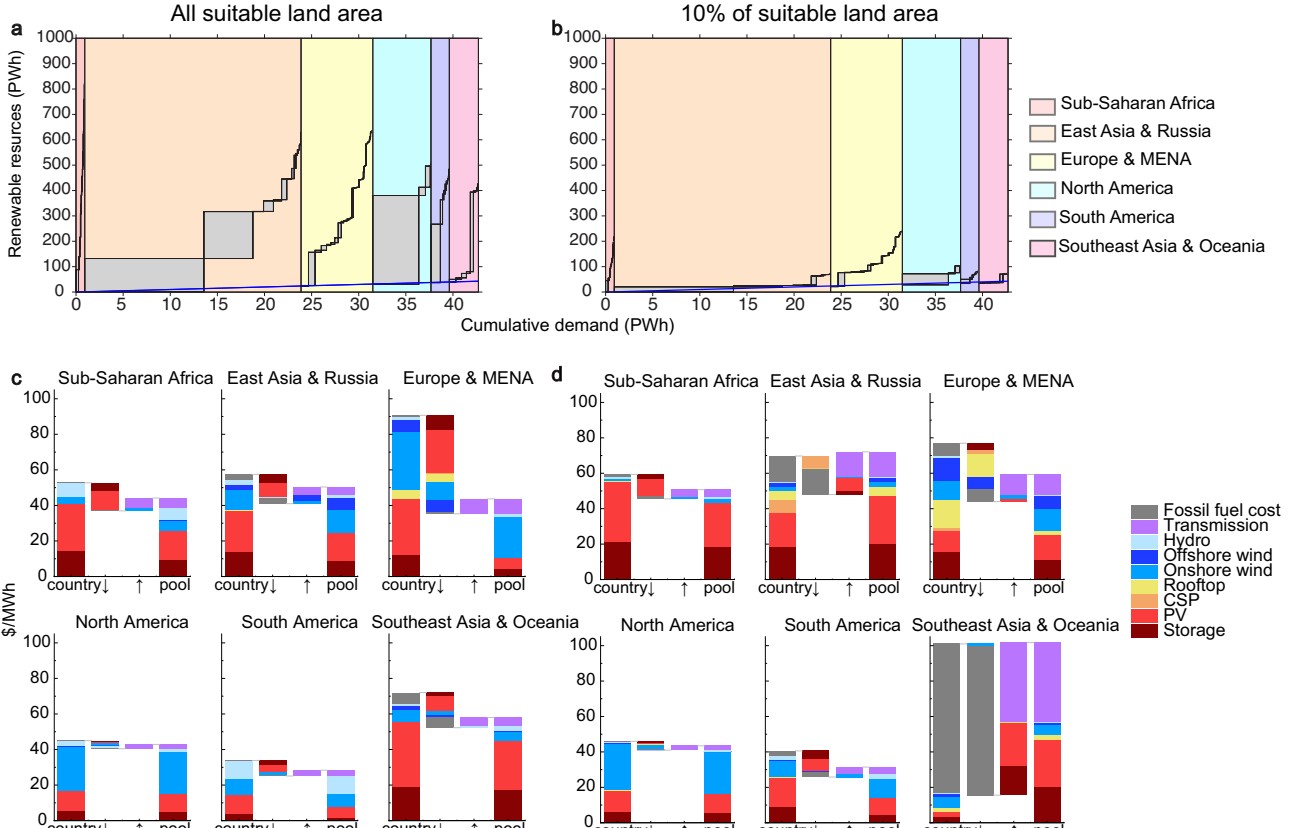

**Fig. 3 | Supply and demand balance of renewable electricity under the 'transcontinental' and 'country' scenarios and the corresponding cost reductions.** **a** Renewable resources with all suitable sites and the 2050 electricity demand by country within continental regions. **b** Renewable resources with top 10% suitable sites and the 2050 electricity demand by country within continental regions. Changes in system cost of electricity ($/MWh) under the transcontinental power pool scenario compared to the 'country' scenario assuming **c** all suitable sites for renewables and **d** only the global top 10% suitable sites are available for development. In **a** and **b**, each rectangle represents electricity demand and renewable resource availability. The blue line represents when cumulative demand equals available renewable resources (y = x line). Within each region, the left-bottom corner of the first rectangle starts at the blue diagonal line because renewable resources are shared only within the continental region.

When utilizing only the top 10% suitable renewable energy sites, benefits of building transcontinental power pools in reducing unmet demand are more pronounced at the regional power pool level (Fig. 3d). Compared to the country scenario, transcontinental power pools reduce unmet demand in East Asia & Russia from 2.7 PWh (12% of regional demand) to 0; in Southeast Asia and Oceania from 2.0 PWh (65%) to 0; and in Europe and MENA from 0.4 PWh (6%) to 0.

Restricting renewable energy development to the top 10% suitable sites results in high system costs in the country scenario. In this case, transcontinental power pools can enable a substantial reduction in system costs, especially in Europe and MENA, South America, and Sub-Saharan Africa because of the development of higher quality renewable energy sites, better balancing of the variability of renewable energy and electricity demand through international trade, and a reduced need for fossil fuel generation to meet electricity demand unmet by renewables. In Europe and MENA, compared to the country scenario assuming no prior interconnections, the transcontinental power pool reduces system cost by 23% because of fewer installed capacities of renewable energy and battery storage, and lower demand for fossil fuel generation. If existing interconnections are considered, this benefit is reduced to 21% but is still large (Supplementary Table 2). The transcontinental power pool scenario also reduces system costs substantially in South America (23%) and Sub-Saharan Africa (14%) compared to the country scenario. The benefits of a transcontinental power pool are modest in North America (6%) because the countries are mostly large in size, providing sufficient high-quality renewable

resources within the country boundaries. System cost increases from transcontinental power pools in East Asia and Russia and Southeast Asia and Oceania are modest−3% and 0.7%, respectively−because decreases in fossil fuel costs and renewable energy are offset by large increases in transmission interconnection costs.

For robustness checks, we also modeled a scenario with only the top 25% suitable sites available for development (Supplementary Fig. 3 and Supplementary Fig. 4) as well as a scenario with 76 PWh of electricity demand in 2050 (~75% greater than the reference scenarios). For all land availability scenarios (10%, 25%, and all), we find that transcontinental power pools meet 100% of global demand with renewables and reduce overall system costs. With the higher 2050 electricity demand, renewables also meet nearly 100% of demand and reduce system costs in all power pools, except that in East Asia and Russia and Southeast Asia and Oceania, the increase in renewable and transmission investment outweighs the decrease in fossil fuel costs when 10% of suitable land is used.

Transcontinental power pools enable electricity trade where countries endowed with inexpensive and abundant renewable resources export electricity to countries with poor endowments of renewable resources (Fig. 4a, b). Using all suitable sites for renewables, the annual trade of electricity reaches ~16% of global demand (Fig. 4a). In Europe and MENA, the annual trade of electricity accounts for nearly 40% of electricity demand. Syria and Oman are the two largest net exporters, followed by Spain and France, all net exporting over 0.2 PWh in a year (Fig. 4a, c). Germany is the largest net importers, net

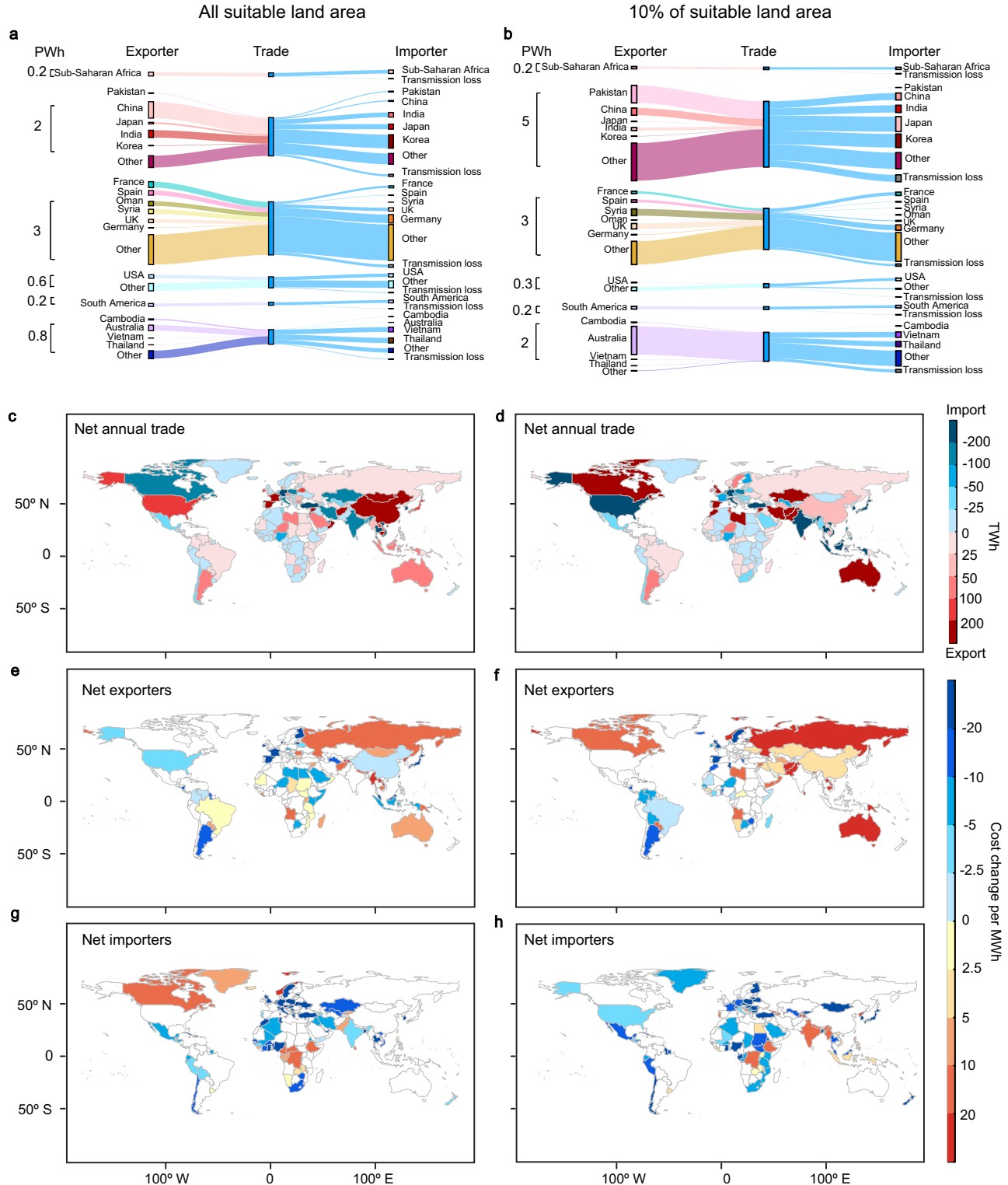

**Fig. 4 | Electricity trade and change in electricity costs under the 'transcontinental' scenario compared to the 'country' scenario.** Annual imports and exports of renewable electricity assuming **a** all suitable sites for renewables, and **b** global top 10% suitable sites are available for development. Annual net export of electricity assuming **c** all suitable sites, and **d** global top 10% suitable sites are available for development. Change of system cost in net exporters assuming **e** all suitable sites for renewables, and **f** global top 10% suitable sites are available for development. Change of system cost in net importers assuming **g** all suitable sites for renewables, and **h** top 10% suitable sites are available for development.

importing nearly 0.5 PWh. In other power pools, the share of traded electricity is over 20% of demand for Sub-Saharan Africa and Southeast Asia and Oceania, and ~10% for East Asia and Russia, North America and South America. Globally, the largest net importer and net exporter both occur in East Asia and Russia: China is the largest net

exporter (0.9 PWh), while South Korea is the largest net importer (0.7 PWh).

By utilizing the top 10% suitable sites, the continental trade of electricity plays a more dominant role, reaching ~30% of the global demand (Fig. 4b). In Southeast Asia and Oceania, the annual trade of

electricity reaches three-quarters (2 PWh) of the regional demand, and nearly all of the net imported electricity is sourced from Australia (Fig. 4b, d), and destined to Indonesia (0.5 PWh), Vietnam (0.5 PWh), Thailand (0.4 PWh) and Malaysia (0.4 PWh). The annual trade of electricity is the largest in East Asia & Russia, reaching 5 PWh, which contributes to a quarter of the regional demand. Specifically, South Korea (1 PWh), India (1 PWh), and Japan (0.6 PWh) are the largest net importers of renewable electricity, while Paksitan (1.5 PWh), Iran (1.3 PWh), Afghanistan (0.8 PWh) and Kazakhstan (0.3 PWh) are the largest net exporters. In Europe and MENA, imported electricity accounts for over 40% (3 PWh) of the regional demand. The largest net exporters are Syria, United Kingdom, Morocco and Libya, aggregately net exporting ~1.5 PWh of electricity for regional trade; large net importers are distributed in Western (e.g., Germany) and Southern Europe (e.g., Italy), benefiting from the inexpensive electricity from the Northern Europe and MENA. In North America, Canada is the largest net exporters (0.3 PWh), while the USA is the largest net importer (0.2 PWh).

To estimate the country-level net benefits or costs of transcontinental power pools compared to the country scenario, we assume that the generation, storage, and transmission costs in a power pool are shared by all countries, proportional to each country's electricity demand. By exporting and importing electricity with neighboring countries, in several net exporters, the transcontinental power pools reduce system costs by reducing generation curtailment, compared to the country scenario (Fig. 4e, f). In both scenarios of land constraints—all suitable sites and only the top 10% suitable sites available for renewable energy development—about a quarter of ~80 net exporters enjoy over $10/MWh reduction in system cost. These benefits in cost reduction are largest in Europe where net exporters experience over $20/MWh decline in system cost.

However, system costs increase in about half of the net exporters because they install more renewable energy capacities and share the cost of electricity generation and transmission (Supplementary Fig. 5) with the net importers within their transcontinental power pool. For example, when utilizing the top 10% suitable sites, the system costs increase by over $20/MWh in Pakistan and Australia because they export large amounts of electricity and share the cost with other members of their power pool (Fig. 4b).

When either all or the top 10% suitable renewable energy sites are available for development, about three-quarters of the net importing countries reduce $5/MWh of the system cost by sharing renewable resources within transcontinental power pools (Fig. 4g, h). Again, net importing countries in Europe experience some of the largest reductions in system costs of electricity (>$20/MWh).

Several importers also see an increase in system costs in transcontinental power pools compared to the country scenario (Fig. 2a, b) when net importers share the system costs with the net exporters in the power pool. Yet, the total system cost of the power pool decreases (Fig. 3c, d) because international trade reduces generation curtailment in the net exporters. For example, the system cost in Canada, a net importer when all suitable sites are used, increases, but the curtailment in the United States, a net exporter, is reduced by 1.3 PWh depending on land constraints.

## Discussion

To achieve net zero greenhouse gas emissions by 2050, it is critical to decarbonize the electricity sector by replacing fossil fuels with renewables. Power pools can reduce costs and help accelerate the phase-out of fossil fuels. Existing examples of multi-country power pools include the Southern African Power Pool, Eastern Africa Power Pool, and Nord Pool[36]. Grid integration projects such as Medgrid[37] and North Seas Energy Cooperaton[38] have been launched to integrate renewable resources in North Europe, North Africa, and the Middle East.

Constraints on land use because of conservation, food production, and other uses can restrict the amount of land required for developing renewable energy. As shown in this study, renewable energy resources can be insufficient to meet all electricity demand within some countries in the absence of international trade, especially with greater constraints on land. Poor endowments of renewable resources, e.g., in Japan and South Korea, can result in high electricity costs and hinder the low-carbon transition in the electricity sector. Transcontinental power pools can not only enable most countries to meet their electricity demand through international trade but also substantially reduce electricity costs by developing the most suitable and least expensive renewable energy sites.

The cross-boundary trade of renewable electricity indicates a new landscape in the global energy market. Historically, fossil fuel resources have also been unequally distributed across countries, and the international trade of fossil fuels has enabled huge profits for exporters of fossil fuels. Building transcontinental power pools is likely to benefit both importers and exporters of renewable energy. By importing electricity, nearly all net importers reduce domestic investments in expensive renewables and storage (Supplementary Fig. 6). By reducing curtailment (Supplementary Fig. 7), about half of the exporters decrease their domestic investment. In this study, we assume that all countries within each transcontinental power pool share the costs in a regional wholesale electricity market. Other cost allocation models such as allocating the lowest-cost renewable resources to consumers within exporting countries and then selling higher-cost resources to importing countries within power pools can change the distribution of system costs across importers and exporters. How to design pricing mechanisms for the transcontinental power pool market remains an open question. The new mechanism needs to equitably allocate the profits from the trade of the electricity market, especially when transmission lines span across several countries.

Geopolitics would be a barrier to building a transcontinental power pool[19]. Creating transcontinental power pools will require a large-scale integration of regional transmission infrastructure, and thus has many challenges including grid ownership, stakeholder roles, financial responsibilities, and revenue allocation between participating countries[39]. Collaboration between countries will be critical in addressing these challenges. Common policy frameworks and agreements need to be reached between national governments[40]. Regional electricity markets and pricing mechanisms need to be established to coordinate between system operators across territories to facilitate power pool operations[41]. The electricity markets of the transcontinental power pools are required to provide a win-win trading mechanism for both exporters and importers.

In the capacity expansion model, we managed to use a time-step of 3 h in a whole year to capture the variation of weather in all 365 days. We find that using 3-h temporal resolution ensures resource adequacy. By using the simulated capacities from the capacity expansion model, the load curtailment of the transcontinental power pool is zero in the hourly operation model (Supplementary Table 3). Following the state-of-the-art, by picking 24 representative days (one peak demand day and one average demand day in a month) and adding 15% planning reserves margins, the transcontinental power pools incurred ~1% load curtailment in the hourly operation model (Supplementary Table 4). Using all 365 days in the capacity expansion model provides more robust quantification results in system costs (Supplementary Table 5) and reliability of the electricity system.

Our research focused on addressing the low-carbon electricity with existing technologies that are commercially mature. Long-term storage technologies (i.e. hydrogen) which could balance the seasonal variability of renewables, are not included in our technologies. Green hydrogen has become a promising long-term storage alternative, but whether green hydrogen can be put into large-scale commercial use is under debate[42,43]. Country-level data on underground storage capacity

for hydrogen is also lacking. Using synthetic methane as the long-term storage is technically mature by using existing infrastructure for natural gas[44,45], but the round-trip efficiency is ~30%[46]. Without the power pools, incorporating long-term storage into our system could reduce the load curtailment by balancing the seasonal variability of renewables, but the transcontinental power pools are still able to reduce the system cost after the inclusion of the long-term storage[47]. Furthermore, under a land-constrained scenario, long-term storage is unable to address the demand shortage caused by the local shortage of renewable energy resources without the international trade of electricity. If renewable energy resources are sourced only within the country boundary, 4.1 PWh of demand shortage still exists due to a shortage of aggregate renewable energy resources when only the top 10% suitable sites are utilized (Supplementary Fig. 1). Our research proves that, without long-term storage, 100% renewable electricity is reliable and economically feasible by expanding transmission lines within continents.

Due to data limitations, our study used uniform cost projections for renewable and storage technologies for all countries and regions. However, costs can vary across countries and regions for various reasons including differences in the cost of capital, access to technologies, and availability of skilled labor. Future analysis could incorporate these cost variations.

Land-use factors for renewables have significant uncertainties given the competing uses of land for agriculture, conservation, and other needs. For example, some sources report land use factors that are double our assumptions adopted from NREL[48,49]. Furthermore, we assumed that only the technology with the least levelized cost can be installed at a particular site. Co-locating renewable energy technologies like wind and solar PV can increase the area of suitable sites and the overall renewable potential.

Last, the cost of transmission in our study was conservatively estimated and may well be an overestimation. We assumed that the cost for HVDC (high-voltage direct current) transmission lines would remain at the 2020 level across the timeframe, but that the transmission cost would decrease due to technology learning. Given these potential overestimations for the costs of transmission lines, the actual economics of transcontinental power pools may be more favorable than how it is portrayed in our study.

## Methods
In the method section, we first measured the development potential of renewable resources. Then we simulated the investment and operation of renewable capacities under the country scenario and the transcontinental scenario, and estimated the costs of electricity to meet the electricity demand in 2050.

### Development potential
We used a published database to quantify the development potential for renewable energy at a 0.01° × 0.01° resolution[16]. Multiple factors including resource yield, market accessibility, population density, infrastructure condition, and land cover were considered to measure the development potential of solar, wind, and hydroelectricity,

$$DPI = w_1 \cdot RY + w_2 \cdot DEG + w_3 \cdot DMR + w_4 \cdot LC \\ + w_5 \cdot DUA + w_6 \cdot DRP + w_7 \cdot IPD \quad (1)$$

where DPI is the development potential index; $w$ is the relative weight for each criterion (Table 1); RY is the resource yield, which is measured as the capacity factor; DEG is the distance to electrical grid; DMR is the distance to major roads; LC is the land cover; DUA is the distance to the urban area; DRP is the distance to railway or ports; IPD is the inverse of the population density.

The relative weights were derived using the analytical hierarchy process. The development potential was classified into 6 levels: very high (top 10%), high (top 10–25%), medium high

**Table 1 | Relative weights for the criteria used in the development potential index**

|  | RY | DEG | DMR | LC | DUA | DRP | IPD |
|---|---|---|---|---|---|---|---|
| CSP | 0.451 | 0.241 | 0.119 | 0.109 | 0.051 | 0.029 |  |
| PV | 0.447 | 0.228 | 0.115 | 0.115 | 0.069 | 0.026 |  |
| Wind | 0.422 | 0.23 | 0.109 | 0.065 | 0.023 | 0.109 | 0.042 |
| Hydropower | 0.453 | 0.11 | 0.051 |  | 0.051 | 0.026 | 0.309 |

(top 25–50%), medium low (50–bottom 25%), low (bottom 25–10%) and very low (bottom 10%). More detailed information and data can be found in Oaklief et al.[16].

The Oaklief database does not include offshore wind and rooftop PV. For the offshore wind, we used the levelized cost to measure the development potential. We only considered the offshore areas which are within 200 kilometers of the coast. We also excluded the protected areas[50], sea ice[51], and areas where wind speeds are less than 8 m/s. For rooftop solar, all the available urban areas were considered to have a very high potential.

### Potential of renewable resources
In our main scenario, for each type of renewable energy technology, we chose (a) all suitable renewable energy sites, and (b) renewable energy sites with very high development potential (global top 10% suitable sites).

In a pixel $x$, we chose the renewable technology with the least levelized cost if multiple technologies are available. The capacity of renewable energy technology $t$ in a pixel $x$ ($C_{t,x}$) was calculated as,

$$C_{t,x} = l_t \cdot A_x \quad (2)$$

where $A_x$ (km²) is the area of the pixel $x$, $l_{t,x}$ (MW/km²) is the land use factor for renewable energy technology $t$.

The capacity of renewable energy $t$ ($C_t$) in a country equals the summation of the renewable capacity in each pixel ($C_{t,x}$) within the country.

$$C_t = \sum_x C_{t,x} \quad (3)$$

$g_{t,x}$ is the maximum electricity generation in a year (8760 h) for renewable energy $t$ in the pixel $x$,

$$g_{t,x} = r_{t,x} \cdot C_{t,x} \cdot 8760 \quad (4)$$

where $r_{t,x}$ is the capacity factor of renewable energy technology $t$ in the pixel $x$.

$g_t$, the maximum generation of renewable energy $t$ in a country, is the summation of electricity generation in pixels within a country,

$$g_t = \sum_x g_{t,x}. \quad (5)$$

$r_t$, the annual average capacity factor of renewable energy technology $t$ in a country, was calculated as,

$$r_t = \frac{g_t}{C_t \cdot 8760} \quad (6)$$

The land use factors were compiled from NREL[48,49,52]. As the land use factor reported in NREL is measured in alternating current power capacity per km², the land use factor for PV power plants was converted between the alternating current and direct current power

capacity by a factor of 1.17[49] (Supplementary Table 6). The annual hydroelectricity generation per pixel was compiled from Hoes et al.[11].

On average, 138 MW rooftop PV ($L_{rooftop}$) can be installed per square kilometer (direct area)[52]. The land use factor for rooftop PV ($l_{rooftop}$) is discounted by the ratio of rooftop area in 1 km² urban area[53] ($\mu$) and the share of suitable rooftop areas ($s$)[12] (Supplementary Table 7).

$$l_{rooftop} = L_{rooftop} \cdot \mu \cdot s \tag{7}$$

Following the method in Wu et al. [12], the capacity factor for solar PV in a pixel ($r_{PV,x}$) was calculated as the ratio of the global titled irradiation (W/m²) in a pixel and the peak solar density of 1000 W/m², adjusted for efficiency losses,

$$r_{PV,x} = \frac{GTI_x \cdot (1 - \eta_0)(1 - \eta_i)}{1000} \tag{8}$$

where $GTI_x$ (W/m²) is the global irradiation for optimally tilted surface in pixel $x$, and was collected from the Global Solar Atlas;[54] $\eta_0$ is the outage rate; $\eta_i$ is the inverter and AC wiring efficiency losses. $\eta_0$ and $\eta_i$ were collected from Wu et al.[35].

The capacity factor of concentrated solar power with no storage in a pixel ($r_{CSP,x}$) was calculated following the empirical linear relationship between the capacity factor and direct normal irradiation[12],

$$r_{CSP,x} = 22.293 \ln(DNI_x) - 145.69 \tag{9}$$

where $DNI_x$ is the direct normal irradiation at pixel $x$ and was collected from the Global Solar Atlas, which derives the average annual capacity factor using daily data from 1994 to 2017[54].

The capacity factor for wind power in a pixel ($r_{wind,x}$) was calculated based on the wind speed,

$$r_{wind,x} = \begin{cases} r_{IEC class I,x}, wind\ speed > 8.5m/s \\ r_{IEC class II,x}, 7.5m/s < wind\ speed \leq 8.5m/s \\ r_{IEC class III,x}, wind\ speed \leq 7.5m/s \end{cases} \tag{10}$$

where $r_{IEC\ class\ I,x}$ is the capacity factor for IEC class I wind turbines; $r_{IEC\ class\ II,x}$ is the capacity factor for IEC class II wind turbines; $r_{IEC\ class\ III,x}$ is the IEC class III wind turbines. The average annual capacity factors and wind speeds were collected from the Global Wind Atlas[55] using 2008–2017 data.

The capacity factor for hydropower in a pixel ($r_{hydro,x}$) and the generation of hydropower were derived from Hoes et al.[11].

$$r_{hydro,x} = 0.5 \tag{11}$$

## Electricity planning model

We used the GridPath model[56] (https://github.com/blue-marble/gridpath), an open-source power system model, to optimize the capacity and generation of energy infrastructure (hydropower, solar, wind, storage, and transmission) in 2050, following a least-cost principle. We first optimized the capacity investment with only renewables and storage by using a 3-h temporal resolution within a whole year, and then simulated the operation with fixed capacities across 8760 h. In the optimization model, the penalty cost for the loss of load is $100 million/MWh. The loss of load in the model is exogenously met by fossil fuel power plants. The levelized cost of electricity includes the cost of renewable energy, the cost of fossil fuels and the social cost of carbon.

The existing capacities for solar, wind, and pumped hydro storage were collected from EIA[57]. The storage duration for the pumped hydro storage was assumed to be 10 hours. Under the transcontinental scenario, we included the existing transmission lines in Europe from Brinkerink et al.[58]. We derived the capital and O&M costs (Supplementary

Tables 6, 8, and 9) and the cost projections (2020–2050, Supplementary Tables 10 and 11) from International Renewable Energy Agency (IRENA)[5] and 2022 Annual Technology Baseline (ATB) by National Renewable Energy Laboratory (NREL)[59]. We collected the transmission loss and cost for HVDC transmission lines (Supplementary Table 9) from Bogdanov et al. The length of a transmission line between countries is the distance of the population centroid between countries[60].

The capacity of the renewable potential derived above is treated as the maximum capacity for new renewable capacities in each country. The planning-reserve margin is 15% of the peak load for each country. We assumed that the maximum discharge duration is 24 h for the battery storage, and that the battery storage is only used to balance supply and demand within a day. The monetary values are all undiscounted 2050 values. Under the country scenario, no transmission lines between countries are built, and under the transcontinental scenario, HVDC transmission lines are built to link all countries within the region (Supplementary Data 1).

Hourly profiles for solar and wind, and load demand were derived from Tong et al.[15], which provides hourly profiles for wind and solar and load in 2018 for 42 major countries and 23 subregions. For the countries that are not included in the 42 major countries, we assumed that their generation and load profiles follow the regional profile. For hydropower, we used a monthly profile derived from IEA monthly data[61] (2015–2021). For countries without hydropower data, we assumed the monthly profile is the same as nearby countries. As there is no IEA data for African countries, we assumed the monthly capacity factor to be 0.5.

The hourly generation profile for solar and wind, and the monthly profile for the hydropower were derived as follows:

$$r_{t,h,m} = \frac{r_t}{r_t^{2018}} \cdot r_{t,h,m}^{2018} \tag{12}$$

$r_{t,h,m}$ is the capacity factor for renewable energy technology $t$ at hour $h$, month $m$; $r_t^{2018}$ is the annual average capacity factor in 2018 from Tong et al.; $r_{t,h,m}^{2018}$ is the capacity factor for renewable energy technology $t$ in 2018 at hour $h$, month $m$.

## Demand scenarios

In the main scenario, we assumed that the growth rates of electricity demand during 2030–2050 follow the IEA projection under the Sustainable Development Goal (SDG scenario). In the sensitivity test, we assumed that the growth rates of electricity demand follow the IEA projection under the Net Zero 2050 scenario (NZE scenario)[62]. The electricity demand in the year 2018 was collected from the Energy Information Administration (EIA)[13]. The growth rates for the two scenarios can be found in Supplementary Table 12.

## Country scenario

The unmet electricity demand by renewables ($short_n$) in the country $n$ under the country scenario was calculated as the difference of load demand ($load_n$) and renewable electricity production ($production\_re_n$),

$$short_n = load_n - production_{re_n} \tag{13}$$

The total cost of electricity demand in the country $n$ under the country scenario ($cost\_country_n$) is the aggregation of renewable electricity cost ($cost\_re_n$) and fossil fuel electricity cost ($cost\_fossil_n$) in the country $n$,

$$cost\_country_n = cost\_re_n + cost\_fossil_n \tag{14}$$

The unmet demand is assumed to be met by fossil fuels. The fossil fuel electricity cost ($cost\_fossil_n$) includes the cost of fossil fuel power

plants and the carbon tax,

$$\text{cost\_fossil}_n = (\overline{\text{lcoe\_fossil}}_n + scc \bullet \overline{ci}_n) \bullet \text{short}_n \qquad (15)$$

where $\overline{\text{lcoe\_fossil}}_n$ is the generation-weighted average cost of fossil fuels in country $n$; $\overline{ci}_n$ is the generation-weighted average $CO_2$ intensity of fossil fuels (ton/MWh) in the country $n$. $scc$ is the social cost of carbon (i.e., carbon tax), which is \$81/tonne $CO_2$ in 2050[29] under a 3% social discount rate. The coal-gas generation mix was derived from world bank[63], and the cost is 95\$/MWh for coal and 90 \$/MWh for natural gas[5,64]; the $CO_2$ intensity for coal and natural gas was collected from NREL ATB.

The system cost of electricity in the country $n$ under the country scenario ($\text{lcoe\_country}_n$) is the cost per load demand,

$$\text{lcoe\_country}_n = \frac{\text{cost\_country}_n}{\text{load}_n} \qquad (16)$$

### Transcontinental scenario

Under the transcontinental scenario, the total cost of electricity in a power pool (total_cost_continent) is the aggregation of renewable electricity cost (total_cost_re), fossil fuel cost (total_cost_fossil), and the transmission cost (total_cost_trans),

$$\text{total\_cost\_continent} = \text{total\_cost\_re} + \text{total\_cost\_fossil} + \text{total\_cost\_trans} \qquad (17)$$

The system cost in the country $n$ ($\text{lcoe\_continent}_n$) is equal to the demand-weighted average system cost in a power pool,

$$\text{lcoe\_continent}_n = \frac{\text{total\_cost\_continent}}{\text{total\_load}} \qquad (18)$$

where total_load is the total load demand in a power pool.

The change of system cost ($\Delta$cost) compared to the country scenario is,

$$\Delta\text{cost} = \text{lcoe\_continent}_n - \text{lcoe\_country}_n \qquad (19)$$

### Data availability

The data used for replicating our analysis are available in the Global Transcontinental Power Pool database under accession code https://doi.org/10.5281/zenodo.10080738. Source Data are provided with the paper. Source data are provided with this paper.

### Code availability

The electricity planning model, GridPath 0.10.1, is available at https://github.com/blue-marble/gridpath. Matlab 2019a and Python 3.8 were used to process the data. Matlab 2019a, Origin 2023 and R 3.6.1 are used for data visualization. All the scripts used in our data collection, data analysis, and data visualization are available at https://github.com/cetlab-ucsb/Transcontinental-power-pool.

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

## Acknowledgements

We acknowledge the support of the Chancellor's Fellowship to Haozhe Yang from UC Santa Barbara. Use was made of computational facilities purchased with funds from the National Science Foundation (CNS-1725797) and administered by the Center for Scientific Computing (CSC). The CSC is supported by the California NanoSystems Institute and the Materials Research Science and Engineering Center (MRSEC; NSF DMR 2308708) at UC Santa Barbara.

## Author contributions

H.Y. curated the data, performed the research, and analyzed the data. R.D. and S.S. supervised the work. H.Y., R.D., and S.S. conceived the idea, designed the study and wrote the manuscript.

## Competing interests

The authors declare no competing interests.
