## [Peer Review File · Nature Communications]

Global transcontinental power pools for low-carbon electricityREVIEWER COMMENTS

Reviewer #1 (Remarks to the Author):

The article "Transcontinental power pools for low-carbon electricity" discusses the important topic – the role of grid interconnections in the energy transition towards higher shares of RE. It is rather obvious that grids will be important for the RE based systems due to various reasons: variability of RE and impact of local weather conditions, uneven distribution of RE resources, area requirements of most of RE technologies which makes impossible local energy supply for densely populated areas with energy intensive economies. Grids reinforcements and further development of transmission grids networks seems inevitable, in some cases for energy transmission from RE rich areas to areas with limited RE resources, or for temporal balancing of supply and demand while most of demand is still covered with local resources. Still it is not clear what are the exact benefits of grids in each of the cases, some researchers are getting more and more skeptical about the concept of super-grids to transfer substantial volumes of electricity over long distances and see the future of more self-sustainable energy system benefiting from energy balancing with neighboring countries' energy systems.

Thought the topic is important, some shortcomings in methods explanations, results presentation and conclusions justifications have to be resolved prior to publications. Also additional language check is strongly advised to improve readability of the text and avoid some disturbing typos.

Here are the detailed comments on the manuscript:

1. Some disturbing typos are still in the text like 'Renewable resurces', see figure 1a y-axis title.
2. Some strange factual mistakes like 'We assume a 4% transmission loss per kilometer' (page 2, line55). Obviously, transmission losses cannot be that high, 4% per 1000 km would be more realistic, but I could not find this number in the referenced paper either.
3. The statement 'By using hourly supply-demand projections and high-resolution renewable resource maps, we found that renewables alone may fall short in meeting global electricity demand in 2050 by 13% without international electricity trade' seems to be incorrect. The 13% shortage is the results of the applied assumptions on 10% of best places used, with different assumption outcome would be different. Even system optimization in full hourly resolution instead of 12 days time-slices would result in higher share of RE as generation and storage capacities would be optimized to reduce fossils use (due to high cost of fuels and CO2 pricing). I suggest authors to reconsider this strong statement to highlight the role of assumptions and methods.
4. Installation density (table s1) and consequently generation potential seems to be underestimated, PV or wind land use factors can be easily twice as much, it can be seen from the real project data. Of course, the authors used the numbers from the referenced source, but reality has to be reflected at least in the text.
5. Why do you consider 10% only and do you consider co-use of area? So if some pixel is included do you only consider PV or Wind potential, or total potential of PV and wind within this pixel?
6. Figure 1a should be better explained. Also it would be valuable to somehow reflect the RE resources actually used to cover demand and what is fossil supply, it will make easier to follow how 5.7 PWh shortage is accumulated.
7. Figure 1a, RE resources of Russia seem to be underestimated, it is counter intuitive that such a big country has such small potential, what is the reason? And What are the numbers for Canada?
8. Figure 1b. What is the reason of perfect match of supply and demand for China? And why countries like USA, Canada, Russia with equally wide area and extended from east to west for thousands of km (should help reduce overall RE variability) cannot balance the demand without fossils?
9. Page 7 line 169, what is the reason of fossil fuels reduction cost?
10. Figure 4a. Please differentiate the colors of supply side flows, so it will be easier to see the source of imported energy.
11. Discussion: it is hard to agree with the given conclusions. The results of the study are strongly affected by the 10% of potential use assumption and strongly distorted by the applied 2-stage modelling in reduced and then full temporal resolution. Authors implemented Top 25% scenario which

leads to improved results in country case, but this scenario highlights the limitation of the 2 stage modeling approach. See table S8, Transcontinental 'Top 10% sites + SDG' leads to 1 PWh shortage, but Transcontinental 'Top 25% sites + SDG' with higher available RE potential and same demand leads to 1.4 PWh shortage. That clearly points on the methods issues which makes impossible to draw proper conclusions based on given results.

12. Methods: Eq 14-15 *penaltyp* is not per MWh it seems to be absolute value

13. Supplementary: formatting to be aligned, now different fonts and different font sizes are used in the tables, sometimes different fonts within one table. Fonts to be harmonized.

I would strongly advise to increase temporal resolution of optimization part it should be possible as technologies portfolio is not that big. The results in higher resolution will be more reliable and then it will be possible to discuss the impact of grids on the RE supply shortage, right now it is hindered by the method inaccuracy.

Reviewer #2 (Remarks to the Author):

The authors submitted a manuscript on the interesting topic of cross-border electricity transmission to support the transition to highly renewable electricity supply. The topic itself is quite relevant and covers frequently discussed issues with renewable energy supply.

While the findings themselves (that electricity transmission can substantially help in reducing the stress for a renewable electricity supply by spreading the generation geographically) are not particularly new and are rather a commonly accepted fact, the quantification of the effects on a global scale with high detail would yield interesting findings and provide significant new results.

However, the study has multiple major shortcomings that would need to be addressed before this manuscript can be recommended for publication:

General remarks:

First, the majority of the results seem to stem from the assumption of using only the top 10% of renewable potentials per country. You show this quite clearly when looking at the graphs in your supplementary material: the unmet demand decreases substantially, especially in those regions where you also find the highest positive effects for the power pools. I do not quite understand why you would make the share of used renewable potential exogenous, instead of using it as an output of your model. In reality, this 10% limitation would not hold, and especially smaller countries or those with a high ratio of energy demand to available area would struggle quite heavily (as can also be seen in your results). What is the reasoning for only allowing them to use their top 10% of potentials? Instead, if it were endogenously optimized, the model could chose the appropriate cost-optimal usage of the available renewable potentials.

Then, and only then, can you check against additional positive effects of the transcontinental power pools, essentially going beyond what can reasonably be achieved within the single country.

You list missing long-term storages as a limitation in your paper, and I see this as a significant issue, since the availability of said storages could change the outcome quite drastically. At the very least, already installed pumped hydro storages or reservoirs should be included in the analysis, wherever they exist.

Also, your methodology section of the paper is written quite sloppily, up to the point where understanding your equations is unclear. There are several typos and wrong idices used in your formulations, making the comprehension quite challenging. Some detailed examples can be found below.

There should also be an extensive spelling and language check, as there are quite several issues within the manuscript.

Other comments:

Abstract:

L9: I disagree with the way you portray your finding here. As stated above, this comes from your 10% assumption and should be very clearly communicated.

Introduction:

L56: Where do you get the 4% transmission loss per km from? I checked the source material and in the supplementary material that you cite, there is a listing of 1.6% of power loss per 1000(!)km for HVDC and 9.4% per 1000km for HVAC lines. Either way, this seems like a typo at the very least, because 4% per km would be extremely excessive.

L62: I do not like the way that you reference many graphs that are not within the manuscript, but in the supplementary material. Either include them in the manuscript, or, if the word/page limit does not allow for it, mention that you have a supplementary material with additional information on the topic, but especially for vital information like your choice of regions, I would rather include it immediately, as this is crucial information that should be easily available to the reader.

L68: I have already given an exhaustive statement as to why I dislike your approach above - but in any case, there should be some reasoning of why you chose this arbitrary value. Why 10%? Why 25%? Also, the results of these sensitivities should be more prominent in the paper, as they put your results in a different perspective.

Results:

L85-90: Again, these results are purely a product of your set limitations. There are also multiple studies out there that show 100% renewable energy supply being possible within these regions, so demand shortages of 10-20% for Europe as an example, seem vastly overstated.

Figure 2: Some countries are grey. What does this mean? Is there no data entry for these countries?

Figure 4: Your grouping of countries within the Sankey diagram is quite confusing. I understand that these kind of graphs can be quickly overwhelming, but having China move into "other" on the demand side, or in North America everything just being "other" seems a bit too aggregated.

L246: Why would exporting countries not have no cost savings in a transcontinental power pool? Given the fluctuating nature of renewables, it might very well be that there are times in a year where also a net-exporting country can benefit from imports from other countries.

Discussion:

L289: Again, this comes down to your set limitation on renewable potentials. Therefore, I think it is wrong to simply dismiss long-term storage like that, just because the chosen overall potential is highly insufficient.

L301: I see this as another major issue. Especially Europe sees 10-20% unmet demand as per your results, and also quite high system costs. This makes sense, given your exclusion of existing grids, but of course makes your results quite meaningless. Especially in Europe, where there are several smaller countries, usually also quite densely populated (and thus quite disadvantaged by the design of your

study), the inclusion of existing transmission would most likely change your results significantly.

Materials and methods:

Table 1: Why is the weight for distance to urban area (DUA) for wind the same as for all the others, or even lower than PV? In wind, especially onshore wind, there are usually quite heavy discussions about local support and minimum distances to settlements, where you would not have this for PV installations, for example.

L353: What do you mean with the land use factor for PV that was converted between AC and DC? Your table S1 lists cost data for technologies and does not have the value 1.17 anywhere in it.

L385: 407: There are multiple typos and missing indices. You list $r(t,h,m)$, but only explain $r(t,h)$, which does not exist in your equation, also $r(t)$ is not explained anywhere.

L428: The description seems to be completely wrong. The equation calculated a generation difference, not costs.

L435: In this equation, you have an addition and a multiplication in the same place. What is correct?

L456: You use the names `cost_pool` and `pool_cost`, which is already confusing enough, but additionally, the indices are once again not matching between equation and description.

Supplementary material:

You need(!) to add sources to your tables. Even if you list the source at some point in the text part of the paper, the sources for data points need to be clearly stated next to each table.

Table S1: The land use factor, is this indeed MW/km²? Why would this be lower for utility-scale PV than for rooftop solar? Also, the number for wind seems low. Please state your assumptions / sources.

Color code:

Comments received are marked in black.

Our responses are marked in blue.

Green texts are the original text in the previous version of the manuscript.

Red texts are new additions.

Line number:

Manuscript with the tracked changes.

Reviewer #1 (Remarks to the Author):

The article “Transcontinental power pools for low-carbon electricity” discusses the important topic – the role of grid interconnections in the energy transition towards higher shares of RE. It is rather obvious that grids will be important for the RE based systems due to various reasons: variability of RE and impact of local weather conditions, uneven distribution of RE resources, area requirements of most of RE technologies which makes impossible local energy supply for densely populated areas with energy intensive economies. Grids reinforcements and further development of transmission grids networks seems inevitable, in some cases for energy transmission from RE rich areas to areas with limited RE resources, or for temporal balancing of supply and demand while most of demand is still covered with local resources. Still it is not clear what are the exact benefits of grids in each of the cases, some researchers are getting more and more skeptical about the concept of super-grids to transfer substantial volumes of electricity over long distances and see the future of more self-sustainable energy system benefiting from energy balancing with neighboring countries’ energy systems.

Thought the topic is important, some shortcomings in methods explanations, results presentation and conclusions justifications have to be resolved prior to publications. Also additional language check is strongly advised to improve readability of the text and avoid some disturbing typos.

1. Some disturbing typos are still in the text like ‘Renewable resurces’, see figure 1a y-axis title.

We apologize for the disturbing typos in the manuscript. We have done a thorough language check to avoid such typos and improve the readability.

2. Some strange factual mistakes like ‘We assume a 4% transmission loss per kilometer’ (page 2, line55). Obviously, transmission losses cannot be that high, 4% per 1000 km would be more realistic, but I could not find this number in the referenced paper either.

We apologize for this typo. The transmission loss is 1.6% per 1000 kilometers (Table S3). We listed the sources for this number in lines 73-74.

3. The statement ‘By using hourly supply-demand projections and high-resolution renewable resource maps, we found that renewables alone may fall short in meeting global electricity demand in 2050 by 13% without international electricity trade’ seems to be incorrect. The 13% shortage is the results of the applied assumptions on 10% of best places used, with different assumption outcome would be different. Even system optimization in full hourly resolution instead of 12 days time-slices would result in higher share of RE as generation and storage capacities would be optimized to reduce fossils use (due to high cost of fuels and CO2 pricing). I suggest authors to reconsider this strong statement to highlight the role of assumptions and methods.

We agree with the reviewer’s comments.

In our revision, we made several improvements.

First, we modeled the power system with

- a) 10% suitable sites and
 - b) all suitable sites for renewable resources,
- and we explained this in lines 84-93,

To understand the implications of land constraints on renewable energy potential, costs, and benefits of transcontinental power pools, we examine two cases of land availability for renewable energy siting. First, all suitable sites for renewables are available for development. However, not all potential renewable resources can be tapped for electricity generation due to constraints on land availability not captured by available geospatial datasets, including ecological impacts, market accessibility, and local political support^{16,34}. Therefore, in the second case, for each renewable energy technology (excluding rooftop PV), we assume that only the top 10% of suitable sites at the global level¹⁶ are available for energy development. We rank and select the top 10% renewable resources based on a composite index of resource yield (annual capacity factor), land use, infrastructure, and market accessibility (see Method section).

Second, as stated in lines 68-69 and lines 496-500, we increased our temporal resolution from 12 days (12·24 hours) to 24 days (24·24 hours). In each month, we chose a representative day on which the average daily load is the closest to the monthly average load; we choose a peak day on which the hourly peak load is the highest.

Lines 68-69

Using the electricity system model, we co-optimize the investment and operation of electricity generation, transmission and storage using 2 representative days per month (24 days).

Lines 496-500

We first optimized the capacity investment with only renewables and storage by using 24 representative days (576 hours) within each year, and then simulated the operation with fixed capacities across 8760 hours. In each month, we chose an ‘average’ day on which the average daily load is the closest to the monthly average load; we choose a ‘peak’ day on which the hourly peak load is the highest.

Last, we reframed our statement and specified the assumptions and methods in lines 11-13.

Under land constraints, if only the global top 10% of suitable renewable energy sites are available, then without international trade, renewables are unable to meet 12% of global demand in 2050.

4. Installation density (table s1) and consequently generation potential seems to be underestimated, PV or wind land use factors can be easily twice as much, it can be seen from the real project data. Of course, the authors used the numbers from the referenced source, but reality has to be reflected at least in the text.

We agree that there is large uncertainty in land use factors.

In our revision, first, we discussed that the land use factor in the real world could be higher than the referenced numbers in Lines 392-397.

Fourth, land-use factors for renewables have significant uncertainties given the competing uses of land for agriculture, conservation, and other needs. For example, some sources report land use factors that are double our assumptions adopted from NREL^{45,46}. Furthermore, we assumed that only the technology with the least levelized cost can be installed at a particular site. Co-locating renewable energy technologies like wind and solar PV can increase the area of suitable sites and the overall renewable potential.

Second, in lines 84-86, we added a case where all available sites are used for electricity generation. In this case, the supply of renewable resources is not a binding constraint to our model.

To understand the implications of land constraints on renewable energy potential, costs, and benefits of transcontinental power pools, we examine two cases of land availability for renewable energy siting. First, all suitable sites for renewables are available for development.

5. Why do you consider 10% only and do you consider co-use of area? So if some pixel is included do you only consider PV or Wind potential, or total potential of PV and wind within this pixel?

We consider the top 10% suitable sites because we want to examine whether the renewables can meet electricity demand with a constraint of land. We discussed why we chose the top 10% suitable lands in the response to the comment 3.

We didn't consider the co-use of the area. This leads to an underestimation of renewable resources under the 10% scenario. We discussed the assumption on co-use of area in lines 395-397.

Furthermore, we assumed that only the technology with the least levelized cost can be installed at a particular site. Co-locating renewable energy technologies like wind and solar PV can increase the area of suitable sites and the overall renewable potential.

In lines 438-439, we explained how we chose the renewable potential within the same pixel.

In a pixel x , we chose the renewable technology with the least levelized cost if multiple technologies are available.

6. Figure 1a should be better explained. Also it would be valuable to somehow reflect the RE resources actually used to cover demand and what is fossil supply, it will make easier to follow how 5.7 PWh shortage is accumulated.

We add a red line in Figure 1a and b to represent the demand met by renewable resources, please see the figure captions in the revised Figure 1 a and 1b.

Figure 1. Supply and demand of renewable electricity in 2050.

Renewable generation and electricity demand in 2050 by country under the 'country' scenario assuming **a** all suitable sites for renewables and **b** top 10% of suitable sites at the global level. Unmet demand with only the supply of renewable resources under the 'country' scenario, assuming **c** all suitable sites for renewables and **d** top 10% of suitable sites at the global level. In **a** and **b**, each rectangle represents the demand for electricity (horizontal dimension) and the available renewable resources potential (vertical dimension). The blue diagonal line represents demand equals renewable electricity potential by country ($y=x$ line). The red line represents the cumulative demand met by renewable electricity generation by

country. Within a country's rectangle, when the slope of the red line is smaller than the slope of the blue line, country-level electricity demand remains unmet using only renewable energy supply.

7. Figure 1a, RE resources of Russia seem to be underestimated, it is counter intuitive that such a big country has such small potential, what is the reason? And What are the numbers for Canada?

Under the 100% land scenario, the number of renewable resources for Russia (41 PWh) and Canada (32 PWh) is lower than China (129 PWh) and USA (349 PWh) despite its large areas. Several reasons account for the lower numbers in Russia and Canada:

1. We exclude the areas where the wind speed is lower than 6 m/s.
2. We exclude the area where the land cover is wetlands, rock or ice.

A more detailed map for available renewable resources is shown in Oaklief et al. 2019, Table S1.

Under the 10% case, the renewable potential in Russia is 1.4 PWh. This number is relatively lower than other large-area countries because the suitability of the sites in Russia is not within the top 10%. The land cover, distance to urban areas and distance to grids degrade the suitability of the land for renewables. Similarly, in Canada, the renewable resource is 3.7 PWh despite the large area.

8. Figure 1b. What is the reason of perfect match of supply and demand for China? And why countries like USA, Canada, Russia with equally wide area and extended from east to west for thousands of km (should help reduce overall RE variability) cannot balance the demand without fossils?

In the original manuscript, the differences are caused by our lower temporal resolution.

In our revision, after increasing the temporal resolution to 24 days, we found that when top 10% sites are used, USA, Canada and Russia match their supply and demand, meeting over 99% of their demand with renewables. In China, 1.0% of demand is infeasible to be met by renewables. Please see the following table or source data provided.

	Unmet demand by renewables using 10% suitable land
China	1.0%
USA	0.7%
Russia	0.1%
Canada	0.2%

9. Page 7 line 169, what is the reason of fossil fuels reduction cost?

The reduction in fossil fuel cost is because after the transcontinental power pool is built, more demand are feasible to be met by renewables. Therefore, less fossil fuels are needed. We explained the reason in lines 237-240.

In Europe & MENA, compared to the 'country' scenario assuming no prior interconnections, the transcontinental power pool reduces system cost by 32% because of fewer installed capacities of renewable energy and battery storage, and lower demand for fossil fuel generation.

10. Figure 4a. Please differentiate the colors of supply side flows, so it will be easier to see the source of imported energy.

In our revision, we show the importing and exporting countries, and differentiate the color on the supply side.

11. Discussion: it is hard to agree with the given conclusions. The results of the study are strongly affected by the 10% of potential use assumption and strongly distorted by the applied 2-stage modelling in reduced and then full temporal resolution. Authors implemented Top 25% scenario which leads to improved results in country case, but this scenario highlights the limitation of the 2 stage modeling approach. See table S8, Transcontinental 'Top 10% sites + SDG' leads to 1 PWh shortage, but Transcontinental 'Top 25% sites + SDG' with higher available RE potential and same demand leads to 1.4 PWh shortage. That clearly points on the methods issues which makes impossible to draw proper conclusions based on given results.

We agree with the reviewer's comments and increased our temporal resolution.

Our two-stage modeling approach is necessary because of computational limitations. As discussed in the methods section, the first stage "capacity investment" model co-optimizes generation and storage investments and operations using representative days and hours whereas the second stage "system operations" model simulates the entire year at an hourly resolution to ensure that the investments made in the first stage are able to reliably meet demand.

Because the representative days in the first stage model may not capture all the peak hours when supply is unable to meet demand, the full-year systems operations simulation can result in more unmet demand. The greater

the number of representative days (and hours) in the first-stage model, the better the model results represent the second-stage model results.

In this revision, we increased the number of sampled days from 12 (288 hours) to 24 (576 hours). The demand shortage results from the first stage are more similar to the second stage (8760-hour simulation) compared to the earlier version of the model. Although the demand shortage for the second stage model for the ‘transcontinental’ scenario still increases slightly from the Top 10% sites to Top 100% sites scenarios, mainly because of the representative day selection issue, the differences are small. More importantly, the key result of demand shortage decreasing in the "transcontinental" scenario compared to the "country" scenario is consistent across all site availability and demand scenarios.

Demand shortage in 2050 under different scenarios of renewable sites and demand. The 2050 global demand is 43 PWh under the SDG scenario and 76 PWh under the NZE scenario.

Scenario		Top 10% sites + SDG	Top 25% sites + SDG	Top 100% sites + SDG	Top 10% sites + NZE	Top 25% sites + NZE	Top 100% sites + NZE
‘Country’	24 days	5.3	1.5	0.8	21	4.1	2.0
	8760 hours	5.3	1.7	1.0	21	4.3	2.4
‘Transcontinental’	24 days	0	0	0	0	0	0
	8760 hours	0.3	0.3	0.4	0.5	0.6	0.7

In the revision, we discussed the simulation error caused by the choice of temporal resolution in lines 368-377.

First, due to the limitation of computation resources, we selected 24 representative days in each year—an average and a peak demand day per month—to optimize the investment of renewable and storage capacities. By choosing representative days based only on electricity demand, one limitation is that one may not capture the peak net demand hours when the generation supply needs to balance demand and maintain the reliability of the power system⁴². Using a full year to optimize the investment of renewables and storage achieves a better performance in reliability of capacity planning, but requires substantial computing resources. We find that using 24 representative days to optimize capacity investments achieves similar performance in terms of the reliability of the electricity system, compared with simulating the operation at an 8760-hour resolution (Table S8).

12. Methods: Eq 14-15 *penalty* is not per MWh it seems to be absolute value

We thoroughly rewrite our method section, and the error above is also corrected. Please see Equation (15)

13. Supplementary: formatting to be aligned, now different fonts and different font sizes are used in the tables, sometimes different fonts within one table. Fonts to be harmonized.

We used a uniform font size and font style for our supporting information in the revision.

14. I would strongly advise to increase temporal resolution of optimization part it should be possible as technologies portfolio is not that big. The results in higher resolution will be more reliable and then it will be possible to discuss the impact of grids on the RE supply shortage, right now it is hindered by the method inaccuracy.

We agree with the reviewer's comments.

In our revision, we increased our temporal resolution to 24 days (576 hours) in a year.

More details can be seen in the response to the comment 11.

Reviewer #2 (Remarks to the Author):

The authors submitted a manuscript on the interesting topic of cross-border electricity transmission to support the transition to highly renewable electricity supply. The topic itself is quite relevant and covers frequently discussed issues with renewable energy supply.

While the findings themselves (that electricity transmission can substantially help in reducing the stress for a renewable electricity supply by spreading the generation geographically) are not particularly new and are rather a commonly accepted fact, the quantification of the effects on a global scale with high detail would yield interesting findings and provide significant new results.

However, the study has multiple major shortcomings that would need to be addressed before this manuscript can be recommended for publication:

General remarks:

1. First, the majority of the results seem to stem from the assumption of using only the top 10% of renewable potentials per country. You show this quite clearly when looking at the graphs in your supplementary material: the unmet demand decreases substantially, especially in those regions where you also find the highest positive effects for the power pools. I do not quite understand why you would make the share of used renewable potential exogenous, instead of using it as an output of your model. In reality, this 10% limitation would not hold, and especially smaller countries or those with a high ratio of energy demand to available area would struggle quite

heavily (as can also be seen in your results). What is the reasoning for only allowing them to use their top 10% of potentials? Instead, if it were endogenously optimized, the model could chose the appropriate cost-optimal usage of the available renewable potentials.

Then, and only then, can you check against additional positive effects of the transcontinental power pools, essentially going beyond what can reasonably be achieved within the single country.

We followed the reviewer's comment and examined two cases in lines 84-93:

- a. All suitable land
- b. 10% of suitable land.

To understand the implications of land constraints on renewable energy potential, costs, and benefits of transcontinental power pools, we examine two cases of land availability for renewable energy siting. First, all suitable sites for renewables are available for development. However, not all potential renewable resources can be tapped for electricity generation due to constraints on land availability not captured by available geospatial datasets, including ecological impacts, market accessibility, and local political support^{16,34}. Therefore, in the second case, for each renewable energy technology (excluding rooftop PV), we assume that only the top 10% of suitable sites at the global level¹⁶ are available for energy development. We rank and select the top 10% renewable resources based on a composite index of resource yield (annual capacity factor), land use, infrastructure, and market accessibility (see Method section).

The results are shown in Figs 1-4. In lines 195-206, we stated that using all suitable land can achieve a large reduction in system costs.

By utilizing all suitable sites for renewable resources, regional renewable resources are larger than their electricity demand by 27-1000 times (Fig 3a). In Sub-Saharan Africa, the renewable resources potential (926 PWh) is nearly 1000 times the 2050 electricity demand (0.9 PWh). In East Asia & Russia where the regional demand is the largest across all regions at 23 PWh, the renewable resources potential is 628 PWh or 27 times the electricity demand. By integrating the regional renewable resources through transcontinental power pools, globally, the unmet demand by renewables decreases from 1 PWh (2%) to 0.4 PWh (1%).

Compared to the 'country' scenario, transcontinental power pools decrease system costs of electricity by 13-45% across all regions except for North America (Fig 3b). Cost reductions in North America are small (3%) because of abundant renewable resources in each of the member countries.

2. You list missing long-term storages as a limitation in your paper, and I see this as a significant issue, since the availability of said storages could change the outcome quite drastically. At the very least, already installed pumped hydro storages or reservoirs should be included in the analysis, wherever they exist.

We agree that including long-term storage could reduce demand shortages, especially those resulting from a seasonal mismatch between demand and renewable energy curtailment and not aggregate renewable resource shortfalls. However, long-term duration is expensive. Even if long-term storage is built, transcontinental power pools will still help reduce system costs.

We explain this in lines 378-380.

Second, due to a lack of data on the storage capacity of hydrogen at the country level, we have not considered long-term hydrogen storage in our model, which could balance the seasonal variability of renewables and potentially reduce the overall system cost.

In our revision, we examined the case where 100% suitable land can be used for renewables, and thus provide sufficient renewable resources. In this case, we find that transcontinental power pools could still reduce the system cost by utilizing the cheapest renewable resources in lines 195-206. Adding long-term hydrogen storage would further reduce the cost of power pools by avoiding generation curtailment due to the seasonal variation of renewable energy.

We also agree that pumped hydro storage can help manage the seasonal variability of renewable energy, and we have added the existing pumped hydro storage in our original manuscript in lines 504-505.

The existing capacities for solar, wind and pumped hydro storage were collected from EIA⁴⁷.

Furthermore, as shown in lines 382-387, in the top 10% case, adding hydrogen won't help meet electricity demand if the renewable potential falls short.

Furthermore, under a land-constrained scenario, long-term storage is unable to address the demand shortage caused by the variability of renewable energy without the international trade of electricity. If renewable energy resources are sourced only within the country boundary, 4.1 PWh of demand shortage still exists due to a shortage of aggregate renewable energy resources when only the top 10% suitable sites are utilized (Fig S1).

Last, for hydrogen, we didn't incorporate it in our analysis because the capacities for underground hydrogen storage at the country level are unknown, and commercial use is still under debate. We discussed how this would change the system cost in lines 378-382.

Second, due to a lack of data on the underground storage capacity of hydrogen at the country level, we have not considered long-term hydrogen storage in our model, which could balance the seasonal variability of renewables and potentially reduce the overall system cost. Green hydrogen has become a promising long-term storage alternative, but whether green hydrogen can be put into large-scale commercial use is under debate^{43,44}.

3. Also, your methodology section of the paper is written quite sloppily, up to the point where understanding your equations is unclear. There are several typos and wrong indices used in your formulations, making the comprehension quite challenging. Some detailed examples can be found below.

Thanks for pointing out this problem. We have rewritten our method section for clarity.

4. There should also be an extensive spelling and language check, as there are quite several issues within the manuscript.

Thanks for pointing out this problem. We have completed a spelling and language check in our paper.

Other comments:

Abstract:

5. L9: I disagree with the way you portray your finding here. As stated above, this comes from your 10% assumption and should be very clearly communicated.

We agree with the reviewer's comments.

In our revision, we clearly stated the assumption behind these results in lines 11-16 in the abstract.

Under land constraints, if only the global top 10% of suitable renewable energy sites are available, then without international trade, renewables are unable to meet 12% of global demand in 2050. Introducing transcontinental power pools with the same land constraints, however, enables renewables to meet 99% of future electricity demand, while also reducing costs by 3-31% across power pools.

Introduction:

6. L56: Where do you get the 4% transmission loss per km from? I checked the source material and in the supplementary material that you cite, there is a listing of 1.6% of power loss per 1000(!)km for HVDC and 9.4% per 1000km for HVAC lines. Either way, this seems like a typo at the very least, because 4% per km would be extremely excessive.

Sorry for the typos.

The transmission line loss is 1.6% per 1000 km in lines 73-74.

We assume a 1.6% transmission loss per 1000 kilometers²⁷.

7. L62: I do not like the way that you reference many graphs that are not within the manuscript, but in the supplementary material. Either include them in the manuscript, or, if the word/page limit does not allow for it, mention that you have a supplementary material with additional information on the topic, but especially for vital information like your choice of regions, I would rather include it immediately, as this is crucial information that should be easily available to the reader.

We merged Figure S1 into Figure 2. We also avoided referring to many supporting graphs in the main text. However, we still refer to some supporting figures. This is because, based on the journal policy, these figures cannot be put into supporting information they are not mentioned in the main text.

8. L68: I have already given an exhaustive statement as to why I dislike your approach above - but in any case, there should be some reasoning of why you chose this arbitrary value. Why 10%? Why 25%? Also, the results of these sensitivities should be more prominent in the paper, as they put your results in a different perspective.

We added a case where the land use is endogenous, which is explained in detail in the response to the comment 1.

We show the sensitivity tests in lines 249-256.

For robustness checks, we also modeled a scenario with only the top 25% suitable sites available for development (Fig S2 and Fig S3) as well as a scenario with 76 PWh of electricity demand in 2050 (~75% greater than the reference scenarios). For all land availability scenarios (10%, 25%, and all), we find that transcontinental power pools meet over 99% of global demand with renewables and reduce overall system costs. With the higher 2050 electricity demand, renewables also meet over 99% of demand and reduce system costs in all power pools, except that in East Asia & Russia, the increase in renewable and transmission investment outweighs the decrease in fossil fuel costs when 10% of suitable land is used.

Results:

9. L85-90: Again, these results are purely a product of your set limitations. There are also multiple studies out there that show 100% renewable energy supply being possible within these regions, so demand shortages of 10-20% for Europe as an example, seem vastly overstated.

We agree with the reviewer that the result relies on our assumptions of land use availability.

In our revision, first, we examine the case where 100% suitable land for renewables is available, and thus provides sufficient renewable resources. We find that most countries can meet over 99% of their demand with sufficient renewable resources in lines 101-102.

Furthermore, we find that renewables alone reliably meet over 99% of electricity demand in 80% of the countries.

Second, we improved the temporal resolution to 24 days in our model, and found that, if 10% suitable sites are used, the demand shortage is over 10% mainly in southern Europe. The shortage will be reduced to 5-10% if existing transmission lines are considered in lines 128-130.

Adding existing inter-country transmission lines reduces the gap in demand and renewable energy supply because of electricity trade, e.g., unmet demand in southern European countries decreases from 10-20% to 5-10% (Fig S7).

10. Figure 2: Some countries are grey. What does this mean? Is there no data entry for these countries?

The countries in grey represent countries with no data. We have explained this in the figure legend in Figure 1 and Figure 2.

11. Figure 4: Your grouping of countries within the Sankey diagram is quite confusing. I understand that these kind of graphs can be quickly overwhelming, but having China move into "other" on the demand side, or in North America everything just being "other" seems a bit too aggregated.

We disaggregated the groups into specific countries, including China and USA.

We revised Fig 4a and Fig 4b. We show the traded electricity, and the importing and exporting countries.

We hope this change can help readers clearly see the source of the exported and imported energy.

12. L246: Why would exporting countries not have no cost savings in a transcontinental power pool? Given the fluctuating nature of renewables, it might very well be that there are times in a year where also a net-exporting country can benefit from imports from other countries.

In our original manuscript, the change of system cost in net exporters is zero because we assumed that the system cost in net exporters equals the cost at the ‘country’ scenario, and the additional cost for exporters (which could be negative) are all absorbed by importers, as shown by equation (18) in the original manuscript.

In our revision, we described how we measure the system cost of net exporters in lines 287-292 and 565-574, which could capture the benefits in cost reduction among net exporters.

Lines 287-292

To estimate the country-level net benefits or costs of transcontinental power pools compared to the ‘country’ scenario, we assume that the generation, storage and transmission costs in a power pool are shared by all countries, proportional to each country’s electricity demand. By exporting and importing electricity with neighboring countries, in several net exporters, the transcontinental power pools reduce system costs by reducing generation curtailment, compared to the ‘country’ scenario (Fig 4c,e).

Lines 565-574

Under the ‘transcontinental’ scenario, the total cost of electricity in a power pool ($total_cost_continent$) is the aggregation of renewable electricity cost ($total_cost_re$), fossil fuel cost ($total_cost_fossil$), and the transmission cost ($total_cost_trans$),

$$total_cost_continent = total_cost_re + total_cost_fossil + total_cost_trans \quad (17)$$

The system cost in the country n ($lcoe_continent_n$) is equal to the demand-weighted average system cost in a power pool,

$$lcoe_continent_n = \frac{total_cost_continent}{total_load} \quad (18)$$

where $total_load$ is the total load demand in a power pool.

The change of system cost ($\Delta cost$) compared to the ‘country’ scenario is,

$$\Delta cost = lcoe_continent_n - lcoe_country_n \quad (19)$$

We explained how the choice of cost allocation could change our results in lines 349-357.

In this study, we assume that all countries within each transcontinental power pool share the costs in a regional wholesale electricity market. Other cost allocation models such as allocating the lowest-cost renewable resources to consumers within exporting countries and then selling higher-cost resources to importing countries within power pools can change the distribution of system costs across importers and exporters. How to design pricing mechanisms for the transcontinental power pool market remains an open question. The new mechanism needs to equitably allocate the profits from the trade of the electricity market, especially when transmission lines span across several countries.

Discussion:

13. L289: Again, this comes down to your set limitation on renewable potentials. Therefore, I think it is wrong to simply dismiss long-term storage like that, just because the chosen overall potential is highly insufficient.

We address this issue by examining the case where 100% suitable land are available for renewables, and thus provides sufficient renewable resources.

More detailed details can be found in the response to the comments 1 and 2.

14. L301: I see this as another major issue. Especially Europe sees 10-20% unmet demand as per your results, and also quite high system costs. This makes sense, given your exclusion of existing grids, but of course makes your results quite meaningless. Especially in Europe, where there are several smaller countries, usually also quite densely populated (and thus quite disadvantaged by the design of your study), the inclusion of existing transmission would most likely change your results significantly.

In our revision, we explained in lines 505-507 that we added the existing transmission lines in Europe under the ‘transcontinental’ scenario.

Under the ‘transcontinental’ scenario, we include the existing transmission lines in Europe from Brinkerink et al⁵³.

In lines 128-130, we stated that we found a reduction in unmet demand after adding the existing transmission lines in Europe, under the ‘country’ scenario.

Adding existing inter-country transmission lines reduces the gap in demand and renewable energy supply because of electricity trade, e.g., unmet demand in southern European countries decreases from 10-20% to 5-10% (Fig S7).

Figure S7. Unmet demand with only the supply of renewable resources under the ‘country’ scenario after adding existing transmission lines in Europe.

a all suitable sites for renewables and **b** top 10% suitable sites at the global level.

In lines 240-241, we stated that after adding transmission lines, power pools still incur a large reduction in system cost.

If existing interconnections are considered, this benefit is reduced to 25% but is still large.

Detailed information can be found in Table S9.

Table S9. System cost (\$/MWh) in Europe & MENA under the ‘country’ with and without existing transmission lines, and the system cost under the ‘transcontinental’ scenario.

Sites	No power pool + No transmission lines	No power pool + Existing transmission lines	Power pool
10%	70	63	47

All	68	58	38
-----	----	----	----

Materials and methods:

15. Table 1: Why is the weight for distance to urban area (DUA) for wind the same as for all the others, or even lower than PV? In wind, especially onshore wind, there are usually quite heavy discussions about local support and minimum distances to settlements, where you would not have this for PV installations, for example.

First, we corrected the numbers in Table 1 due to our careless typos.

Second, for wind, the relative weights are derived by comparing the relative importance of the factors for wind per se, not by comparing the factors for wind between solar or other renewable energy.

For wind, population density (IPD) is the proxy for local support and distance to settlements, and is given a higher weight than the distance to urban areas (DUA). In fact, only wind and hydropower consider the population density, while other renewable energy doesn't consider this item.

A more detailed explanation of the weights can be found in the supplementary of Oaklief et al. 2019.

Table 1. Relative weights for the criteria used in the development potential index.

	RY	DEG	DMR	LC	DUA	DRP	IPD
CSP	0.451	0.241	0.119	0.109	0.051	0.029	
PV	0.447	0.228	0.115	0.115	0.069	0.026	
Wind	0.422	0.23	0.109	0.065	0.023	0.109	0.042
Hydropower	0.453	0.11	0.051		0.051	0.026	0.309

16. L353: What do you mean with the land use factor for PV that was converted between AC and DC? Your table S1 lists cost data for technologies and does not have the value 1.17 anywhere in it.

In NREL, the land use factor is reported as MW (alternating current)/km². Our capacity factor measures the electricity generated per MW (direct current). To keep the numbers consistent, the MWac/km² is converted to MWdc/km² by using a factor of 1.17, which is used by NREL. We show the value of 1.17 in Table S1 and explain this in lines 458-461.

As the land use factor reported in NREL is measured in the alternating current power capacity per km², the land use factor for PV power plants was converted between the alternating current and direct current power capacity by a factor of 1.17 (Table S1).

17.

L385: 407: There are multiple typos and missing indices. You list $r(t,h,m)$, but only explain $r(t,h)$, which does not exist in your equation, also $r(t)$ is not explained anywhere.

We explained the indices in line 530. $r(t)$ is explained in lines 454-456.

L428: The description seems to be completely wrong. The equation calculated a generation difference, not costs.

We corrected this mistake in line 543.

L435: In this equation, you have an addition and a multiplication in the same place. What is correct?

We corrected this typo in equation (15).

L456: You use the names `cost_pool` and `pool_cost`, which is already confusing enough, but additionally, the indices are once again not matching between equation and description.

We rewrote the method in lines 564-574.

Supplementary material:

18. You need(!) to add sources to your tables. Even if you list the source at some point in the text part of the paper, the sources for data points need to be clearly stated next to each table.

In our revision, we added sources of data in all our tables in the supporting information.

19. Table S1: The land use factor, is this indeed MW/km^2 ? Why would this be lower for utility-scale PV than for rooftop solar? Also, the number for wind seems low. Please state your assumptions / sources.

Response

In the original text, we reported the installed rooftop PV capacity per floor area in Table S1, without considering the discount factor including the availability and the suitability of rooftop area in urban areas. In our revision, after using the discount factor, the land use factor for rooftop PV is $11.5 \text{ MW}/\text{km}^2$ urban area. Please check lines 463-465.

On average, 138 MW rooftop PV (L_{rooftop}) can be installed per square kilometer (direct area)⁴¹. The land use factor for rooftop PV (l_{rooftop}) is discounted by the ratio of rooftop area in 1 km^2 urban area⁴² (μ) and the share of suitable rooftop areas (s)¹⁴ (Table S6).

The land use factor for wind is much lower than PV because wind turbines need to be spaced very far apart. The land use factor for wind is sourced from the report of NREL “Land-Use Requirements of Modern Wind Power Plants in the United States”, page 10, which summarized the total area required for building wind farms. Please check the data source in 392-294 and line 458.

Reviewers' comments:

Reviewer #1 (Remarks to the Author):

The reviewers' questions were answered, and the authors tried to improve the methods used in the study in order to comply with the reviewers' requests. However, temporal resolution of 24 days is still too low to properly simulate RE based system operation, especially on global level.

With limited number of time slices researcher has to pick representative days or other periods of time to consider all the variety of conditions observed in the system. For one region it is manageable, but in this study such approach is not applicable: For global study of transmission grids impact, researcher has to simulate energy exchange between regions, so days picked for time slices must be same globally. At the same time, on the global scale the extreme conditions will be observed on different days so different days have to be picked. So we have two conditions, which, considering variety of conditions around the globe, can hardly be managed with only 24 days' time slices. Whole year simulation, in hourly or 2-3 hours resolution would provide more reliable results.

Considering all that, estimation of transcontinental transmission grids impact may be inaccurate. Qualitative conclusion on the importance of power grids to balance VRE supply and energy demand is not affected by the methods flaw, but does not represent any novelty.

1. Have the other reviewers raised technical issues that you feel are important to address? Do you disagree with any of their technical criticisms? Have the authors successfully responded to these requests?

Yes. I agree with all raised points, similar concerns on the used methods of modelling were raised in my comments.

The authors have formally responded to the given requests, and improved the methods especially in part on use of total available potential instead of 10%, however the methods used is still inadequate to reflect the RE based energy system specifics and the quantitative results are questionable.

Qualitative results are not questionable, these are quite obvious and do not content novelty.

2. Do the other reviewers' comments alter your stance on the conceptual advance and/or novelty of the study?

These are R1 comments I've already seen in the response to reviewers. It is fully inline with my view on the study.

I was in serious doubt on the decision for revised manuscript. The authors formally responded the raised comments, so fulfilled the request of reviewers and then the manuscript has to be accepted. At the same time I think that methods are still inadequate and have to be revised. The proper quantification of all aspects will demand higher spatial resolution, inclusion of more technologies (especially storage), higher temporal resolution, etc. Finally the new methods have to be developed.

Overall, after your request and more analysis I tend to request major revision or reject the manuscript.

Responses to the reviewer comments

Color codes:

Comments received.

Our responses.

Original text in the previous version of the manuscript.

New additions.

1. The reviewers' questions were answered, and the authors tried to improve the methods used in the study in order to comply with the reviewers' requests. However, temporal resolution of 24 days is still too low to properly simulate RE based system operation, especially on global level.

With limited number of time slices researcher has to pick representative days or other periods of time to consider all the variety of conditions observed in the system. For one region it is manageable, but in this study such approach is not applicable: For global study of transmission grids impact, researcher has to simulate energy exchange between regions, so days picked for time slices must be same globally. At the same time, on the global scale the extreme conditions will be observed on different days so different days have to be picked. So we have two conditions, which, considering variety of conditions around the globe, can hardly be managed with only 24 days' time slices. Whole year simulation, in hourly or 2-3 hours resolution would provide more reliable results.

Considering all that, estimation of transcontinental transmission grids impact may be inaccurate. Qualitative conclusion on the importance of power grids to balance VRE supply and energy demand is not affected by the methods flaw, but does not represent any novelty.

We thank the reviewer for the constructive comments. The primary concern that Reviewer 1 raised hinges upon the fact that we used 24-day sample data to model a year-long period for capacity investment. Instead, the reviewer noted, advancing the literature would require using 2-3 hour intervals throughout one whole year.

We agree with the reviewer that the use of a 3-hour time-step over all 365 days of the year would improve the reliability of the result. Doing so was technically challenging given the amount of computational resources that it requires. Our previous results already tested the limit of our supercomputing resources that we had access to, and the use of 3-hour time-step throughout the year rather than the sample of 24 days would require an order of magnitude larger computational power and time.

However, following the reviewer's recommendation, **we managed to achieve the 3-hour time-step in 365 days over a whole year** through parallelization of our supercomputing resources (see the lines 69-70 and 545-547). That means that we were able to use $8,760/3 =$

2,720 hours, instead of 576 hours, of sample data to construct our model following the reviewer’s comment. It is important to note, however, that our new results based on 3-hour time-step confirm our previous conclusions, while the numerical results did show minor changes. Without the transcontinental power pools, our new model shows that the unmet demand would amount to 0.8 PWh under 100% land availability scenario and 5.2 PWh under the 10% land availability scenario, while those from the previous model using representative days were 1.0 PWh and 5.3 PWh, respectively (Table S8- S9). With transcontinental power pools, the new model shows that the system costs similarly decrease in most power pools (Table S10).

We believe that our new results more concretely demonstrate the benefits of building power pools as a potential solution for addressing the variability and intermittency of renewable electricity at a global scale in the course of low-carbon transition, while successfully addressing the reviewer's concern.

We explained the advance of our model in lines 399-407.

In the capacity expansion model, we managed to use a time-step of 3 hours in a whole year to capture the variation of weather in all 365 days. We find that using 3-hour temporal resolution ensures resource adequacy. By using the simulated capacities from the capacity expansion model, the load curtailment of the transcontinental power pool is zero in the hourly operation model (Table S8). Following the state-of-the-art, by picking 24 representative days (one peak demand day and one average demand day in a month) and adding 15% planning reserves margins, the transcontinental power pools incurred ~1% load curtailment in the hourly operation model (Table S9). Using all 365 days in the capacity expansion model provides more robust quantification results in system costs (Table S10) and reliability of the electricity system.

Table S8. The unmet demand using the capacity expansion model with a 3-hour temporal resolution model, and the validation using an 8760-hour operation model. The 2050 global demand is 43 PWh under the SDG scenario.

Scenario	Model	Top 10% sites + SDG	Top 100% sites + SDG
‘Country’	3-hour ¹	12%	1.9%
	8760 ²	12%	1.9%

‘Transcontinental’	3-hour	0	0
	8760	0	0

¹ Optimized the capacity investment by using 3-hour temporal resolution in a whole year,

² Simulated the operation with fixed capacities across 8760 hours.

Table S9. The unmet demand using the capacity expansion model with 24 representative days, and the validation using an 8760-hour operation model. The 2050 global demand is 43 PWh under the SDG scenario.

Scenario	Model	Top 10% sites + SDG	Top 100% sites + SDG
‘Country’	Reduced ¹	12%	1.9%
	8760 ²	12%	2.2%
‘Transcontinental’	Reduced	0	0
	8760	0.7%	0.9%

¹ Optimized the capacity investment by using 24 representative days (576 hours) within a year,

² Simulated the operation with fixed capacities across 8760 hours.

Table 10. Change of the system cost in transcontinental power pools compared with the case without power pool by using different temporal resolutions in the capacity expansion model.

Pool	100% land availability		10% land availability	
	24 days	3-hour	24 days	3-hour
Sub-Saharan Africa	-14%	-16%	-11%	-14%
East Asia & Russia	-13%	-12%	-5.6%	3.3%
Europe & Middle East	-44%	-52%	-32%	-23%
North America	-2.9%	-4.7%	-3.4%	-5.5%

South America	-13%	-16%	-19%	-23%
Southeast Asia & Oceania	-15%	-19%	-4.4%	0.7%

1. Have the other reviewers raised technical issues that you feel are important to address? Do you disagree with any of their technical criticisms? Have the authors successfully responded to these requests?

Yes. I agree with all raised points, similar concerns on the used methods of modelling were raised in my comments.

The authors have formally responded to the given requests, and improved the methods especially in part on use of total available potential instead of 10%, however the methods used is still inadequate to reflect the RE based energy system specifics and the quantitative results are questionable.

Qualitative results are not questionable, these are quite obvious and do not content novelty.

We believe that our response above has successfully addressed the reviewer's concern.

2. Do the other reviewers' comments alter your stance on the conceptual advance and/or novelty of the study?

These are R1 comments I've already seen in the response to reviewers. It is fully inline with my view on the study.

I was in serious doubt on the decision for revised manuscript. The authors formally responded the raised comments, so fulfilled the request of reviewers and then the manuscript has to be accepted. At the same time I think that methods are still inadequate and have to be revised. The proper quantification of all aspects will demand higher spatial resolution, inclusion of more technologies (especially storage), higher temporal resolution, etc. Finally the new methods have to be developed.

Overall, after your request and more analysis I tend to request major revision or reject the manuscript.

First, for the spatial resolution, in lines 62-63, we collected the data on renewable resources at a spatial resolution of $0.01^\circ \times 0.01^\circ$, and aggregated these data into the country level. A country-level analysis is by far the finest resolution for a global study to our knowledge.

Second, in our paper, we included pumped hydro storage as the review 2 requested. We didn't include long-term storage like hydrogen for the following reasons, and detailed analysis can be seen in lines 420-435.

1. The focus of our study is addressing low-carbon electricity with existing technologies that are mature and currently being commercially used. Hydrogen and other long-term storage technologies, though promising in the future, have not been commercially applied, and could incur large uncertainties in costs and potential capacities.
2. Our results show that 100% renewable electricity could be achieved at a low cost by expanding transmission lines even without long-term storage like hydrogen, which itself has an important policy implication. This means that 100% electricity system is feasible given existing technologies, and it is unnecessary to delay our climate actions by waiting for the maturity of hydrogen technologies.
3. The inclusion of hydrogen and other long-term storage technologies won't influence our conclusion that building transcontinental power pools can reduce the storage capacity, and thereby reduce the system cost of power pools.
4. Under a land-constrained scenario, global demand is falling short by 4.3 PWh due to a lack of renewable resources, if sourced within country boundaries. Adding long-term storage won't help meet the demand caused by the lack of resources.

Our research focused on addressing the low-carbon electricity with existing technologies that are mature and are commercially used. Therefore, promising long-term storage technologies (i.e. hydrogen) which could balance the seasonal variability of renewables, are not included in our technologies. Green hydrogen has become a promising long-term storage alternative, but whether green hydrogen can be put into large-scale commercial use is under debate^{43,44}. Country-level data on underground storage capacity for hydrogen is also lacking. Furthermore, under a land-constrained scenario, long-term storage is unable to address the demand shortage caused by the local shortage of renewable energy resources without the international trade of electricity. If renewable energy resources are sourced only within the country boundary, 4.1 PWh of demand shortage still exists due to a shortage of aggregate renewable energy resources when only the top 10% suitable sites are utilized (Fig S1). Our research proves that, without long-term storage, 100% renewable electricity

is reliable and economically feasible by expanding transmission lines within continents.

Third, we managed to include all 365 days with a time-step of 3 hours, which satisfied the reviewer's request, as have been answered in the comments above.

REVIEWERS' COMMENTS

Reviewer #1 (Remarks to the Author):

The authors managed to make a substantial improvement in the study methods and simulate the global system for the whole year in 3-hours resolution. Even though authors claim that 'the numerical results show minor changes', the methods improvements lead to significant qualitative change in the results, now the results clearly show that the global energy demand can be covered with renewables without use of fossil fuels.

I suggest to accept the manuscript with minor revisions.

Some minor comments:

1. You claim that the long-term energy storage technologies are not mature, having hydrogen storage as an example. However, there exist other options, namely methane storage. Natural gas storage capacities are present in every country consuming gas these days, and the same capacities and infrastructure can be used to store synthetic methane (SNG). Methane synthesis is also a mature technology with more than 8 bcm/y capacity only considering China and USA. Most of existing capacities use fossil feedstock, but renewables can be used in future. However, round trip efficiency in case of SNG long-term storage would be much lower even compared with hydrogen long-term storage. If long term storage would be integrated the supply and demand would be balanced in many countries, though some countries would still face energy deficit in 'country' scenario as it is claimed by authors.

Please reflect existence of mature long-term storage technologies and its possible impact on results in the limitations section of the Discussion.

2. The Table 10 in supplementary to be changed to Table S10, in accordance to other captions format.

Responses to the reviewer comments

Color codes:

Comments received.

Our responses.

Original text in the previous version of the manuscript.

New additions.

Reviewer #1 (Remarks to the Author):

The authors managed to make a substantial improvement in the study methods and simulate the global system for the whole year in 3-hours resolution. Even though authors claim that ‘the numerical results show minor changes’, the methods improvements lead to significant qualitative change in the results, now the results clearly show that the global energy demand can be covered with renewables without use of fossil fuels.

I suggest to accept the manuscript with minor revisions.

Some minor comments:

1. You claim that the long-term energy storage technologies are not mature, having hydrogen storage as an example. However, there exist other options, namely methane storage. Natural gas storage capacities are present in every country consuming gas these days, and the same capacities and infrastructure can be used to store synthetic methane (SNG). Methane synthesis is also a mature technology with more than 8 bcm/y capacity only considering China and USA. Most of existing capacities use fossil feedstock, but renewables can be used in future. However, round trip efficiency in case of SNG long-term storage would be much lower even compared with hydrogen long-term storage.

If long term storage would be integrated the supply and demand would be balanced in many countries, though some countries would still face energy deficit in ‘country’ scenario as it is claimed by authors.

Please reflect existence of mature long-term storage technologies and its possible impact on results in the limitations section of the Discussion.

We thank the reviewer for pointing out this caveat in our discussion.

In our revision, we stated the existence of mature long-term storage, and the potential impacts on our results. Please see lines 408-418.

Our research focused on addressing the low-carbon electricity with existing technologies that are **commercially** mature. Long-term storage technologies (i.e. hydrogen) which could balance the seasonal variability of renewables, are not included in our technologies. Green hydrogen has become a promising long-term storage alternative, but whether green hydrogen can be put into large-scale commercial use is under debate^{42,43}. Country-level data on underground storage capacity for hydrogen is also lacking. **Using synthetic methane as the long-term storage is technically mature by using existing infrastructure for natural gas^{44,45}, but the round-trip efficiency is ~30%⁴⁶. Without the power pools, incorporating long-term storage into our system**

could reduce the load curtailment by balancing the seasonal variability of renewables, but the transcontinental power pools are still able to reduce the system cost after the inclusion of the long-term storage⁴⁷.

2. The Table 10 in supplementary to be changed to Table S10, in accordance to other captions format.

We changed Table 10 to Table S5, after we numbered the tables based on the order of tables in the main text. Please see Supplementary Information.